# Automated bone marrow cytology using deep learning to generate a histogram of cell types

Rohollah Moosavi Tayebi[1,2], Youqing Mu [1], Taher Dehkharghanian[1], Catherine Ross[1,3], Monalisa Sur[1,3], Ronan Foley[1,3], Hamid R. Tizhoosh[2,4] & Clinton J. V. Campbell [1,3 ✉]

## Abstract

**Background** Bone marrow cytology is required to make a hematological diagnosis, influencing critical clinical decision points in hematology. However, bone marrow cytology is tedious, limited to experienced reference centers and associated with inter-observer variability. This may lead to a delayed or incorrect diagnosis, leaving an unmet need for innovative supporting technologies.

**Methods** We develop an end-to-end deep learning-based system for automated bone marrow cytology. Starting with a bone marrow aspirate digital whole slide image, our system rapidly and automatically detects suitable regions for cytology, and subsequently identifies and classifies all bone marrow cells in each region. This collective cytomorphological information is captured in a representation called Histogram of Cell Types (HCT) quantifying bone marrow cell class probability distribution and acting as a cytological patient fingerprint.

**Results** Our system achieves high accuracy in region detection (0.97 accuracy and 0.99 ROC AUC), and cell detection and cell classification (0.75 mean average precision, 0.78 average F1-score, Log-average miss rate of 0.31).

**Conclusions** HCT has potential to eventually support more efficient and accurate diagnosis in hematology, supporting AI-enabled computational pathology.

## Plain language summary

Identifying and counting cells in bone marrow samples, known as cytology, is critical for the diagnosis of blood disorders. This is a complex and labor-intensive process, with some variation in how hematopathologists interpret these samples. Here, we develop an artificial intelligence system for automated bone marrow cytology, which automatically detects and identifies all types of cells found in the bone marrow. This information is summarized in a chart that we call the Histogram of Cell Types (HCT), a new way to represent complex information generated in bone marrow cytology. Our system achieves high accuracy and precision in classifying the different types of bone marrow cells as a HCT. This tool may eventually help clinicians to make more efficient and accurate diagnoses.

[1] McMaster University, Hamilton, ON, Canada. [2] Kimia Lab, University of Waterloo, Waterloo, ON, Canada. [3] Juravinski Hospital and Cancer Centre, Hamilton, ON, Canada. [4] Artificial Intelligence and Informatics, Mayo Clinic, Rochester, MN, USA. ✉email: campbecj@mcmaster.ca

A bone marrow study is the foundation of making a hematological diagnosis, with an estimated 700 000 bone marrow studies performed annually in the US[1]. It is performed to investigate a clinically suspected hematological disorder, as part of lymphoma staging protocols and to assess bone marrow response to chemotherapy in acute leukemias[2]. Information is extracted by a hematopathologist from the multiple components that comprise a bone marrow study and then integrated with clinical information to make a final diagnostic interpretation[2]. Much of this interpretation relies on visual features of bone marrow cells and tissue viewed through a light microscope[2] or more recently, via high-resolution scanned digital whole slide images (WSIs) of pathology specimens, known as digital pathology[3,4]. One component of a bone marrow study, called the aspirate, consists of particles of bone marrow tissue that are smeared onto a glass slide to allow individual bone marrow cells to be analyzed for subtle and complex cellular features that represent the morphological semantics of the tissue, known as cytology[2,5]. As per international standards, aspirate cytology includes a nucleated differential cell count (NDC), where 300-500 individual bone marrow cells are manually identified, counted, and classified into one of many discrete categories by a highly experienced operator such as a hematopathologist[2]. Bone marrow cytology and the NDC are required for many critical clinical decision points in hematology. For example, the identification of leukemic blasts may lead to immediate initiation of flow cytometry, karyotype, and induction chemotherapy in acute myeloid leukemia (AML)[6,7]. Similarly, the identification of subtle cytological changes in bone marrow cells is necessary for the diagnosis and risk stratification in patients with a myelodysplastic syndrome (MDS)[8]. Failure to recognize and quantify abnormal cell populations in the aspirate in a timely and accurate manner may lead to delayed or incorrect diagnosis. In the context of a busy reference hematopathology lab, performing cytological review on every bone marrow aspirate specimen is tedious and subject to inter-observer variability[9–11]. At the same time, smaller community centers often lack sufficient technical expertise to correctly interpret bone marrow aspirate cytology[12]. One study estimated that up to 12% of MDS cases are misdiagnosed due to the inability to recognize morphological dysplasia in aspirate specimens in less experienced centers[11]. This leaves an unmet clinical need for innovative computational pathology tools that will support the aspirate review process.

Artificial Intelligence (AI) describes the aspiration to build machines, or computer software, with human-like intelligence[13,14]. One particular type of AI algorithm, called deep learning, has shown considerable success in digital image analysis and image classification tasks in many domains[15,16]. In the pathology domain, deep learning represents a computational pathology tool that has been successfully implemented in many non-hematopoietic pathology sub-specialties using WSIs of solid tissue pathology specimens, known as histopathology[17]. Numerous studies have demonstrated the ability of deep networks to perform tasks such as binary morphological classification, distinguishing tumor from normal tissue[18–23], as well as histomorphological tissue grading[24]. While these approaches generally deliver excellent classification results, they do not capture the nuances or complexity inherent in bone marrow aspirate cytology. Specifically, the vast majority of morphological analysis in the hematopoietic system is performed at the level of cellular resolution and represents non-binary classification based on subtle morphological features such as dysplasia in MDS. The application of deep learning to diagnostic hematopathology will therefore require unique solutions that are tailored to these distinct cytomorphological challenges.

While there are several commercial computational pathology workflow support tools developed for analysis of peripheral blood cytology[25], there are currently no clinical-grade solutions available for bone marrow cytology. In comparison to blood film cytology, bone marrow aspirates are complex cytological specimens. Aspirates contain only a small number of regions suitable for cytology, significant non-cellular debris and many different cell types that are often aggregated or overlapping[2,5]. This has rendered bone cytology as a relatively challenging computational pathology problem. Aspirate cytology can be roughly modeled into three distinct computational steps to reflect real-world hematopathology practice. The first problem is region of interest (ROI) detection, where a small number of regions or tiles suitable for cytology must be selected from large WSI prior to cell detection and classification. ROI selection has previously been accomplished in bone marrow aspirates by a human operator manually selecting and cropping the appropriate tiles in aspirate WSIs[26,27]. Second, there is the problem of object detection, where individual bone marrow cells or non-cellular objects must be identified in aspirate WSI as both distinct and separate from background. Prior approaches have employed deep learning for object detection such as regional CNN (R-CNN), fast and Faster R-CNN[28,29]. These approaches utilize region proposals for object detection followed by a separate method such as object classification, which renders them complex to train and hence computationally inefficient[26,30,31]. Third and finally there is the problem of object classification, where individual bone marrow cells or non-cellular objects must be assigned to one of numerous discrete classes based on nuanced and complex cytological features. This complexity increases in MDS, where morphological dysplasia creates subtle cytological changes.

One study attempted to address the second and third problems using fine-tuning of Faster R-CNN and the VGG16 convolutional network[26]. However, this approach proved operationally slow and is not likely scalable to a clinical diagnostic workflow. Therefore, novel, efficient and scalable computational pathology approaches are needed to support bone marrow aspirate cytology; specifically approaches that add full end-to-end automation, i.e., from unprocessed WSI to bone marrow cell counts and classification.

Recently, a deep learning model called You Only Look Once (YOLO) was developed for real-time object detection to specifically address the detection and classification problems in complex image analysis domains[31]. YOLO uniquely allows for object detection and classification to occur in a single step, where all objects in an image are simultaneously identified and localized by a "bounding box" and then assigned a class probability by the same deep network[31]. The YOLO model outputs a set of real numbers that captures both object localization in an image and an object class probability, therefore solving both object detection and classification problems simultaneously in a regression approach[31]. In addition, the most recent version of YOLO, YOLOV4, has been optimized for small object detection and uses complete intersection-over-union loss (CIoU), which results in faster convergence and better accuracy for bounding box prediction[32]. These factors collectively lead to increased computational efficiency and speed compared to previous methods[28–32]. YOLO can perform object detection and classification on multiple image objects which are complex and overlapping in virtual real-time (milliseconds)[31], and consequently has been applied in several real-world problems including autonomous driving[31,33–35]. Recently, YOLO has been applied in some medical domain problem such as pathology. For example in ref. [36], YOLO has been applied to assess the cell types in bone marrow smears. However, only 7 cell types have been considered in that study. Moreover, the tiles need to be selected manually by the user.

In this work, we demonstrate the first automated end-to-end AI architecture for bone marrow aspirate cytology. We first

employ and implement a fine-tuned DenseNet model to rapidly and automatically select appropriate ROI tiles from a WSI for bone marrow aspirate cytology. Subsequently, we implement a YOLO model trained from scratch to detect and assign class probabilities to all cellular and non-cellular objects in bone marrow aspirate digital WSI. Collective cytological information for each patient is then summarized as a Histogram of Cell Types (HCT), which is a novel information summary quantifying the class probability distribution of bone marrow cell types, acting as a cytological fingerprint. A histogram is generally a representation of a distribution, a very old graphical technique to count discrete values[37]. Our approach shows cross-validation accuracy of 0.97 and precision of 0.90 in ROI detection (selecting appropriate tiles), and mAP (mean Average Precision) of 0.75 and average F1-score of 0.78 for detecting and classifying 16 key cellular and non-cellular objects in aspirate WSIs. Our approach has potential to fundamentally change the process of bone marrow aspirate cytology, leading to more efficient, more consistent and automated diagnostic workflows, and providing a foundation for computational pathology driven augmented diagnostics and precision medicine in hematology.

## Methods

This work proposes a new end-to-end AI architecture for bone marrow aspirate NDC based on machine learning and deep learning algorithms (Fig. 1).

**Dataset.** This study was approved by the Hamilton Integrated Research Ethics Board (HiREB), study protocol 7766-C. As this study protocol was retrospective, it was approved with waiver of patient consent. Digital whole slide images (WSI) were acquired retrospectively and de-identified and annotated with only a diagnosis, spanning a period of 1-year and 1247 patients. This starting dataset represented the complete breadth of diagnoses over this period in a major hematology reference center. WSI were then sampled from this dataset for model development and validation as described in Table 1 and Supplementary Table S3. These images were scanned with either an Aperio Scanscope AT Turbo or a Huron TissueScope at 40X and acquired as SVS and tif file format.

**Data annotation and augmentation strategy.** ROI tiles and individual bone marrow cell types included were annotated by expert hematopathologists as the ground truth or reference standard in WSI images used for model training and test-validation as described below. This follows ICSH guidelines, where expert pathologists are considered the reference standard for bone marrow aspirate cytology in clinical diagnosis[2]. Data augmentation was applied to increase the diversity of the input image types. Generally, there are two categories for pixel-wise adjustments augmentation, photometric distortion, which includes hue, contrast, brightness, saturation adjustment, and adding noise; and geometric distortion, which includes flipping, rotating, cropping, and scaling. As we had an imbalanced class

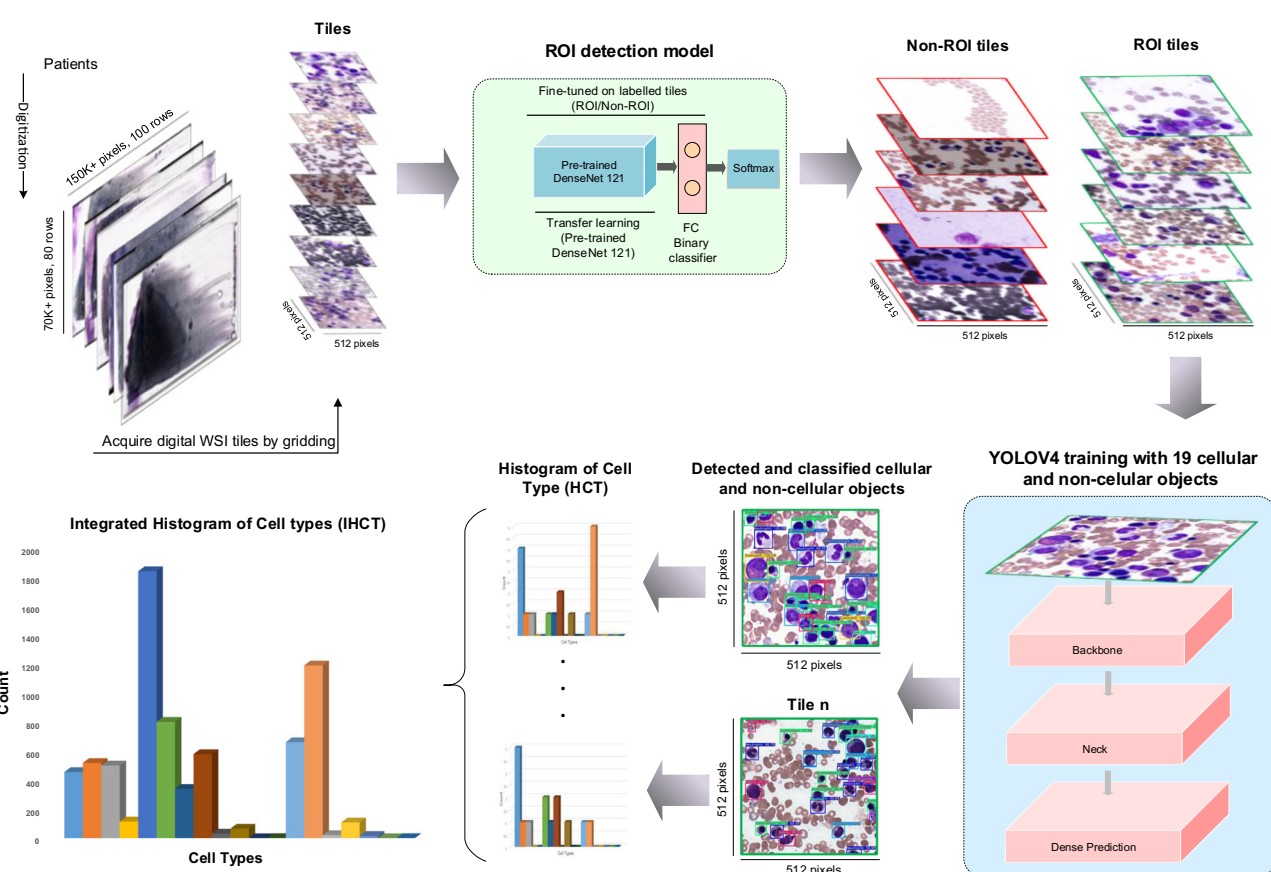

**Fig. 1 End-to-end AI architecture for bone marrow aspirate cytology.** In this architecture, initially, our Region of Interest (ROI) detection model is run on unprocessed bone marrow aspirate WSI. A grid is created on an original Whole-Slide Image (WSI) and ROI tiles are selected using ROI detection model. Subsequently, a You-Only-Look-Once (YOLO)-based object detection and classification is run to localize and classify cells in the selected tiles and generate the Integrated Histogram of Cell Types (IHCT).

**Table 1 Diagnostic tags and the number of patient WSI used for training and test-validation in each category for the ROI detection model.**

| Diagnostic tags | Used in training | Used in test-validation | Number of patients |
|---|---|---|---|
| Normal | 80 | 18 | 98 |
| Myelodysplastic syndrome (MDS) | 15 | 3 | 18 |
| Acute leukemia | 23 | 5 | 28 |
| Lymphoproliferative disorder | 28 | 7 | 35 |
| Plasma cell neoplasm | 19 | 4 | 23 |
| Hypercellular | 5 | 1 | 6 |
| Erythroid hyperplasia | 3 | 0 | 3 |
| Myeloproliferative neoplasm (MPN) | 4 | 1 | 5 |
| Inadequate | 11 | 3 | 14 |
| Hypocellular | 6 | 2 | 8 |
| MPN/MDS | 2 | 0 | 2 |
| MPN | 3 | 1 | 4 |
| Necrosis | 2 | 1 | 3 |
| Carcinoma | 3 | 0 | 3 |
| Total | 204 | 46 | 250 |

expert hematopathologist; appropriate ROI tiles needed to be well spread, thin and free of red cell agglutination, overstaining and debris; and contain at least one segmentable cell or non-cellular object as outlined above. Then, a deep neural network based on DenseNet121 architecture[39] was fine-tuned to extract features from each tile. A binary classifier was added in the last layer of the model to classify appropriate and inappropriate tiles. This network was trained using a cross entropy loss function and AdamW optimizer with learning rate 1e-4 and weight decay 5.0e-4. Also, a pretrained DenseNet121 was applied to initialize all weights in the network prior to fine-tuning. The entire network was fine-tuned for 20 epochs with 32 batch size.

We applied patient-level 5-folds cross-validation to train and test the model. Hence, the dataset (98,750 tiles) was split into two main partitions in each fold, training and test-validation, 80% (204 WSIs including 80,250 tiles) and 20% (46 WSIs including 18,500 tiles), respectively. The test-validation was also been split into two main partitions, 70% validation and 30% test. To ensure that enough data for each class was chosen in our dataset, the above split ratios were enforced on appropriate and inappropriate tiles separately. The dataset was split into training, validation and test sets at patient level, such that each set has a patient WSI that does not come in the other sets to prevent data leakage. In each fold, the best model was picked by running on the validation partition after the training and then evaluated on unseen patients in the test dataset. Extracting ROI tiles for further processing for the cell detection and classification model was the primary aim of using the ROI detection model. To this end, the ROI detection model should be able to minimize false positives in the result. Therefore, the precision has been considered as a key performance metric to select the best model.

distribution within our dataset for the ROI detection model (70,250 inappropriate and 4,750 appropriate tiles), it was necessary to apply one of the over-sampling or under-sampling methods to prevent misclassification. To address this, a number of the above augmentation techniques were applied to the training data during the learning process to over-sample the appropriate ROI tiles and train the model correctly. Subsequently, after applying augmentation, the dataset in this phase contained 98,750 annotated images for training, including 70,250 inappropriate ROI tiles and 28,500 appropriate ROI tiles (Supplementary Table S1). For the cell detection and classification model, after annotating the objects inside ROI tiles by using LabelImg tool (Supplementary Fig. S1)[38], in addition to the above augmentation categories, other techniques were also applied, like cutmix[39] which mixes 2 input images, and mosaic, which mixes 4 different training images. Accordingly, after applying augmentation, the dataset in this phase contained 1,178,408 annotated cells for training, including 119,416 neutrophils, 44,748 metamyelocytes, 52,756 myelocytes, 17,996 promyelocytes, 173,800 blasts, 117,392 erythroblasts, 1,012 megakaryocyte nuclei, 57,420 lymphocytes, 25,036 monocytes, 7,744 plasma cells, 10,956 eosinophils, 308 basophils, 4,664 megakaryocytes, 246,532 debris, 8,404 histiocytes, 1,452 mast cells, 174,724 platelets, 25,740 platelet clumps, and 88,308 Other cell types (Supplementary Table S2). To enhance generalization, the augmentation was only applied on the training set in each fold of the cross-validation.

**Region of interest (ROI) detection method**. The first phase in the proposed architecture is ROI detection. The ROI detection was applied to extract tiles from a WSI and examine if that tile was suitable for diagnostic cytology. To accomplish this, a deep neural network was built, fine-tuned and evaluated on aspirate digital WSI tiles. In the ROI detection method, initially 98,750 tiles (including augmented data) in 512 × 512-pixel size in high resolution are extracted and acquired from 250 WSI. To choose the tiles, a grid of 15 rows and 20 columns was created on each digital WSI and tiles were selected from the center of each grid cell, ensuring all tiles have been sampled from the WSI evenly. Appropriate and inappropriate ROI tiles were annotated by an

**Cell detection and classification**. The next phase was cell detection and classification applied on ROI tiles of 512 × 512 pixels in high resolution. To accomplish this, the YOLOv4 model was customized, trained and evaluated to predict bounding boxes of bone marrow cellular objects (white blood cells) and non-cellular objects inside the input ROI tile and classify them into 19 different classes. In this architecture, CSPDarknet53[40] was used as the backbone of the network to extract features, SPP[41] and PAN[42] were used as the neck of the network to enhance feature expressiveness and robustness, and YOLOv3[43] as the head. As bag of specials (BOS) for the backbone, Mish activation function[44], cross-stage partial connection (CSP) and multi input weighted residual connection (MiWRC) were used. For the detector, Mish activation function, SPP-block, SAM-block, PAN path-aggregation block, and DIoU-NMS[45] were used. As bag of freebies (BoF) for the backbone, CutMix and Mosaic data augmentations, DropBlock regularization[46], and class label smoothing were used. For the detector, complete IoU loss (CIoU-loss)[45], cross mini-Batch Normalization (CmBN), DropBlock regularization, Mosaic data augmentation, self-adversarial training, eliminate grid sensitivity, using multiple anchors for single ground truth, Cosine annealing scheduler[47], optimal hyperparameters and random training shapes were used. In addition, the hyperparameters for bone marrow cell detection and classification were used as follows: max-batches is 130,000; the training steps are 104,000 and 117,000; batch size 64 with subdivision 16; the polynomial decay learning rate scheduling strategy is applied with an initial learning rate of 0.001; the momentum and weight decay are set as 0.949 and 0.0005 respectively; warmup step is 1,000; YOLO network size set to 512 in both height and width; anchor size set to 13, 14, 19, 18, 29, 30, 19, 64, 62, 20, 41, 39, 35, 59, 50, 49, 74, 35, 56, 62, 68, 53, 46, 87, 70, 70, 95, 65, 79, 85, 101, 95, 87,129, 139,121, 216, 223.

Similar to the ROI detection method above, patient-level 5-folds cross-validation was applied to train the model here. Therefore, each fold is divided into training and test-validation partitions, 80% and 20% respectively. The test-validation data portion was split into two main partitions (70% validation and 30% test). Additionally, to ensure that enough data for each class was chosen in our dataset, the mentioned portions were enforced on each object class type individually. In each fold, the best model was picked by running it on the validation partition and then evaluation on the test (unseen) dataset was performed using the mean average precision (mAP).

After training and applying the cell detection and classification model on each tile, the Chi-square distance (Eq. (1)) was applied to determine when the IHCT converges.

$$\tilde{\chi}^2 = \frac{1}{2} \sum_{i=1}^{n} \frac{(x_i - y_i)^2}{(x_i + y_i)} \tag{1}$$

If IHCT converged, the bone marrow NDC is completed and represented by the IHCT, otherwise, another tile is extracted, and the previous process applied again iteratively until it converges.

To calculate Chi-square distance, the number of following cellular objects, as well as $BM_{ME}$ ratio (Eq. (2)), were utilized: "neutrophil", "metamyelocyte", "myelocyte", "promyelocyte", "blast", "erythroblast", "lymphocyte", "monocyte", "plasma cell", "eosinophil", "basophil", "megakaryocyte".

$$BM_{ME} \; ratio$$
$$= \frac{Blast + Promyelocyte + Myelocyte + Metamyelocyte + Neutrophil + Eosinophil}{Erythroblast} \tag{2}$$

Cell types were chosen to include all bone marrow cell types traditionally included in the NDC, as well as several additional cell or object types that have diagnostic relevance in hematology ("megakaryocytes", "megakaryocyte nuclei", "platelets", "platelet clumps" and "histiocytes").

**Evaluation**. To evaluate the ROI detection model in predicting appropriate and inappropriate tiles, we calculated common performance measures such as accuracy, precision (PPV-positive predictive value), recall (sensitivity), specificity, and NPV (negative predictive value), as shown by the following equations:

$$Accuracy = \frac{T_p + T_n}{T_p + T_n + F_p + F_n} \tag{3}$$

$$Precision = \frac{T_p}{T_p + F_p} \tag{4}$$

$$Recall = \frac{T_p}{T_p + F_n} \tag{5}$$

$$Specificity = \frac{T_n}{T_n + F_p} \tag{6}$$

$$NPV = \frac{T_n}{T_n + F_n} \tag{7}$$

For the ROI detection model, $T_p$, $T_n$, $F_p$ and $F_n$ in the above equations are defined as:

$T_p$ (True Positive): The number of appropriate tiles which predicted correctly

$T_n$ (True Negative): The number of inappropriate tiles which predicted correctly

$F_p$ (False Positive): The number of inappropriate tiles which predicted as appropriate tiles

$F_n$ (False Negative): The number of appropriate tiles which predicted as inappropriate tiles

To assess the performance of the proposed cell detection and classification method, Average Precision (AP) was used with 11-point interpolation (Eq. (8)). Also at the end, the mean Average Precision (mAP) [48] was calculated for all the AP values (Eq. (10)). The value of recall was divided from 0 to 1.0 points and the average of maximum precision value was calculated for these 11 values. It is worth mentioning that the value of 0.5 was considered for Intersection over Union (IoU) in AP for each object detection and >0.75 has been used for class probability. In addition, Precision, Recall, F1-score (Eq. (12)), average IoU (Eq. (11)) and log-average miss rate (Eq. (13)) have been calculated here for each object type.

$$AP = \frac{1}{11} \sum_{r \in \{0.0, \dots, 1.0\}} AP_r = \frac{1}{11} \sum_{r \in \{0.0, \dots, 1.0\}} P_{interp}(r) \tag{8}$$

where

$$P_{interp}(r) = \max_{\tilde{r} \geq r} p(\tilde{r}) \tag{9}$$

$$mAP = \frac{1}{N} \sum_{i=1}^{N} AP_i \tag{10}$$

$$IoU = \frac{GTBox \cap PredBox}{GTBox \cup PredBox} \tag{11}$$

$$F1 - Score = 2 \times \frac{Precision \times Recall}{Precision + Recall} \tag{12}$$

For the cell detection and classification model, $T_p$, $F_p$ and $F_n$ in Eq. (4) and Eq. (5) are defined as:

$T_p$ (True Positive): The number of all cellular and non-cellular objects which predicted correctly.

$F_p$ (False Positive) and $F_n$ (False Negative): The number of all cellular and non-cellular objects which not predicted correctly

The Log-average miss rate [49] is calculated by averaging miss rates at 9 evenly spaced FPPI points between $10^{-2}$ and $10^0$ in log-space.

$$Log_{average \; miss \; rate} = \left( \prod_{i=1}^{n} a_i \right)^{\frac{1}{n}} = \exp \left( \frac{1}{n} \sum_{i=1}^{n} \ln a_i \right) \tag{13}$$

Where $a_1, a_2, \dots, a_9$ are positive values corresponding the miss rates at 9 evenly spaced FPPI points in log-space, between $10^{-2}$ and $10^0$.

**Reporting summary**. Further information on research design is available in the Nature Research Reporting Summary linked to this article.

## Results

**Automatic detection of regions suitable for bone marrow cytology**. Following bone marrow biopsy specimen acquisition from a hematology patient, particles of bone marrow tissue are smeared (push preparation) or crushed (crush preparation) onto a glass slide releasing individual bone marrow cells which are then fixed, stained and analyzed by a hematopathologist as described above and in refs. [2,5]. This is called a bone marrow aspirate smear. In digital pathology, glass slides of bone marrow aspirate smears are scanned using a digital slide scanner to generate a high-resolution WSI for a hematopathologist to review. To this end, we sampled digital WSI from our starting dataset of 1247 bone marrow aspirate WSIs acquired over the span of one year at the hematology reference center, Hamilton Health Sciences. This dataset represented the complete breadth of diagnoses

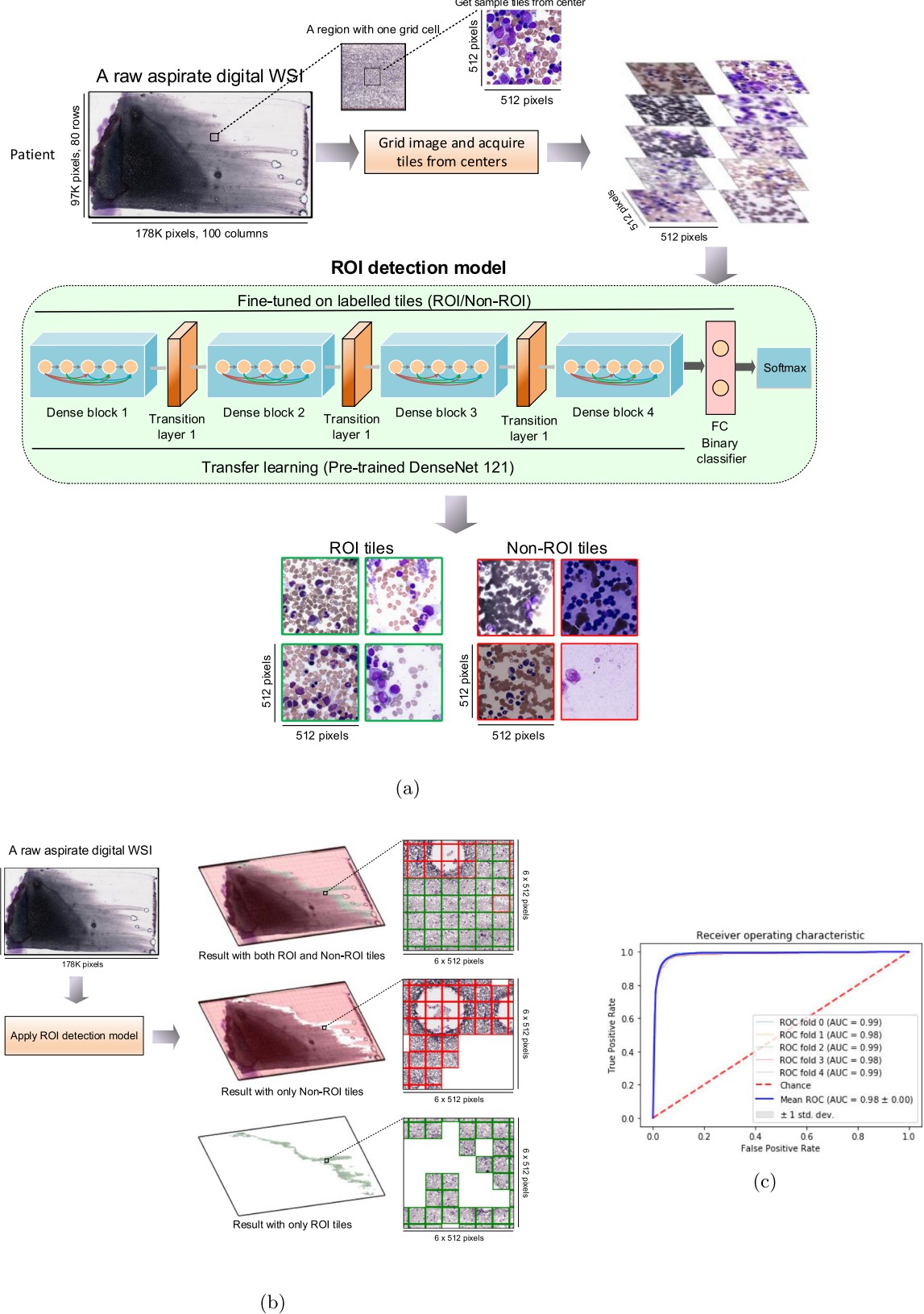

**Fig. 2 Applying the region of interest (ROI) detection model. a** Example of a raw aspirate whole-slide image (WSI), the tile grid to ensure all tiles have been sampled from the WSI evenly, the Region of Interest (ROI) detection model, and output examples of applying the model to separate appropriate tiles from inappropriate tiles. **b** Example of applying the ROI detection model to display the entire ROI and Non-ROI inside an aspirate WSI. **c** Mean Receiver Operating Characteristics (ROC) curve of the ROI detection model. All the results are aggregated over all 5-folds.

**Table 2 Evaluation of the ROI detection using 5-fold cross-validation to calculate accuracy, precision (PPV-Positive Predictive Value), recall (Sensitivity), specificity, and NPV (Negative Predictive Value).**

| Metrics | % |
|---|---|
| Average Cross-validation Accuracy | 0.97 |
| Average Cross-validation Precision (PPV) | 0.90 |
| Average Cross-validation Specificity | 0.99 |
| Average Cross-validation Recall (Sensitivity) | 0.78 |
| Average Cross-validation NPV | 0.99 |

All these metrics were computed in each test (unseen) fold separately and then the average was calculated.

and cytological findings seen over this period (see methods, Table 1 and Supplementary Table S3 for details).

To first address the ROI detection, we developed a deep model to automatically detect regions in bone marrow aspirate WSIs that are suitable for cytology. An aspirate WSI may contain only a small number of regions suitable for cytology; these regions are thinly spread, free from significant cellular overlap and over-staining, and clearly show the subtle and complex cytological features required for cell classification[5]. To this end, we implemented a fine-tuned DenseNet 121 architecture (Fig. 2a) to select and classify individual tiles as ROI tiles (appropriate tiles) and non-ROI tiles (inappropriate tiles) (Fig. 2b and Supplementary Fig. S2). All layers of this model were fine-tuned on over 98,750 tiles of 512 × 512 pixels from bone marrow aspirate WSIs representing 250 patients that were randomly selected from our starting dataset and annotated by expert hematopathologists as ROI or non-ROI (Supplementary Fig. S3 and Table 1). Based on these criteria, the dataset was divided into 28,500 appropriate and 70,250 inappropriate ROI tiles (total of 98,750 tiles including data augmentation) which were then used to train the ROI detection model. The model was then validated by partitioning the data at patient level into training and test-validation sets, in which 46 patients including 18,500 tiles were considered for testing the results in each fold of 5-folds cross-validation. In addition, both crush and push preparation aspirate specimens were included in the training and testing data set to enhance the robustness of the training model across multiple preparation modalities. The ROI detection model was evaluated on imbalanced data (more non-ROI than ROI tiles) in order to reflect a real-world scenario, where only 10-20% of the WSI may be useful for cytology. Results are shown in Table 2 and Fig. 2c; the model achieved accuracy, precision, specificity, recall (sensitivity) and NPV of 0.97, 0.90, 0.99, 0.78 and 0.99, respectively. These findings demonstrated our deep learning ROI detection model was able to automatically select tiles from a bone marrow WSI appropriate for bone marrow cytology with high accuracy and precision, providing the foundation for an automated end-to-end bone marrow cytology model and abrogating the need for manual ROI identification.

**YOLO learning for bone marrow cell detection and classification.** Following the development of our DenseNet ROI detection model, we applied a YOLO model on selected appropriate ROI tiles to automatically detect and classify all bone marrow cellular and non-cellular objects. Here, all cellular and non-cellular objects (excluding red blood cells) in each ROI tile in bone marrow aspirates were detected and assigned a class probability (Fig. 3a and Supplementary Fig. S4). Using ROI tiles selected by our fine-tuned DenseNet model as input, we trained a YOLO model from scratch to detect all bone marrow cell types included

in the NDC (neutrophils, metamyelocytes, myelocytes, promyelocytes, blasts, erythroblasts, lymphocytes, monocytes, plasma cells, eosinophils, basophils, mast cells), in addition to histiocytes, platelets, platelet clumps, megakaryocytes, megakaryocyte nuclei and debris, which are cells and non-cellular objects that are not part of the traditional NDC, but may have specific diagnostic relevance to hematopathology (Fig. 3c).

To facilitate object annotation, we applied our ROI detection model on WSIs from 106 patients randomly selected from our starting dataset to extract appropriate ROI tiles, representing 10 diagnostic categories, the majority of which were diagnostically annotated as abnormal. (Table S3). We then annotated the location of each object with a bounding box, and subsequently each object was assigned to one of the above object classes by an expert hematopathologist. Similar to clinical practice, objects that could not be classified with certainty by a hematopathologist were labeled as *other cells*. Therefore the model would be trained not to assign these to any specific category. While there are clear weaknesses in such an approach, due to the requirements of YOLO model training, leaving cells without an annotation was not possible. The trained model was then validated with approximately 250,000 objects (inside 26,400 ROI tiles) considered for evaluation in each fold of a 5-folds cross-validation (Table 3 and Fig. 3b). The model achieved a high mAP and average F1 score in object detection and classification: mAP, average F1-score, precision and recall are 0.75, 0.78, 0.83, and 0.75, respectively, where the highest classification was achieved for eosinophil and erythroblast with AP 0.97 and 0.92, respectively, while megakaryocyte nucleus and histiocyte showing the most classification errors with AP 0.60 and 0.54, respectively, which may be a result of class imbalances or cytological heterogeneiety of these relatively rare objects. Cell types such as blasts and lymphocytes which may show overlapping morphological features, also showed lower model performance in accuracy, similar to expert human hematopathologists. Model performance in the specific individual diagnostic categories of normal, MDS, acute leukemia, plasma cell neoplasm and lymphoproliferative disorder can be found in Supplementary Fig. S5.

**Improving YOLO model performance using active learning.** Active learning broadly describes numerous ML approaches where data that are either underrepresented or address weaknesses in model performance are queried and then labeled as training data[50]. This allows for generalization of a relatively small amount of labeled training data to a large unlabeled datasets, which is of particular relevance to medical domains such as pathology where well-annotated training data is scarce. To accordingly augment our data set and improve performance and training efficiency of our YOLO model, we designed a unique strategy called active learning. Here, model training started with a relatively small dataset and was then improved iteratively by expert evaluation for weaknesses in performance. In this way, the expected error reduction (EER) approach of active learning was used; at each iteration, wrongly classified cells were re-labeled and added to the current training dataset to train the model in the next iteration. During the first and second training iterations (before implementing active learning), new ROI tiles were fully annotated manually by an expert hematopathologist to train our YOLO model, 719 tiles representing 32 WSI were used for full cell annotation. From the third iteration onward, our active learning approach was employed to annotate 2766 new tiles representing an additional 74 WSIs, which were validated by our team of 4 expert hematopathologists and hematologists with 5–35 years of pathology experience. (Table S3 and Supplementary Table S2).

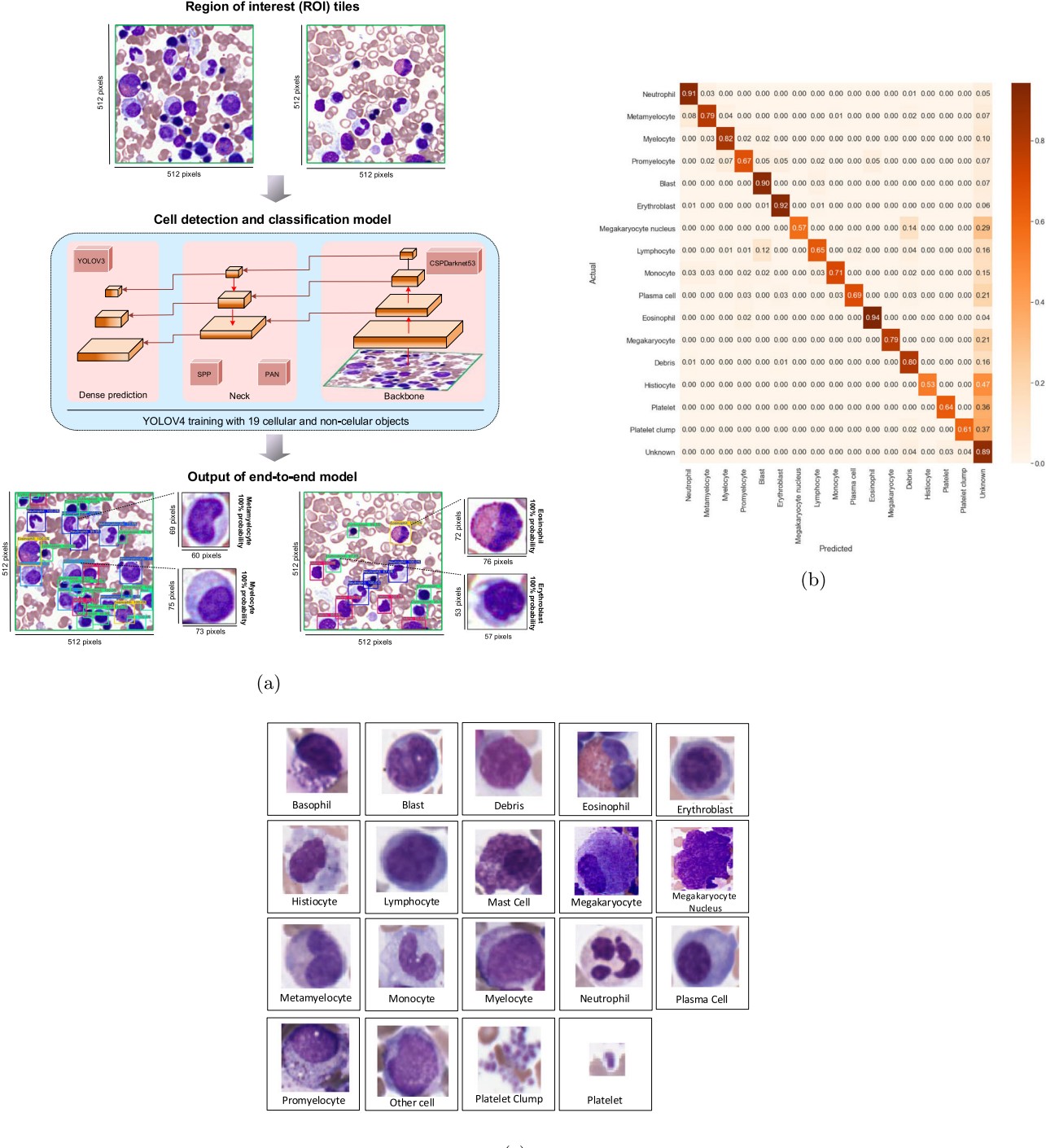

**Fig. 3 Applying the YOLO model to localize objects in selected region of interest (ROI) tiles. a** Example of a Region of Interest (ROI) tile, as the output of the ROI detection model, the You-Only-Look-Once (YOLO) cell detection and classification model architecture, and output examples of applying the YOLO model to detect and classify objects inside the input ROI tiles. **b** The cross-validation confusion matrix showing the performance of the YOLO cell detection and classification model applied on 16 different cytological and non-cytological object types representing 10 diagnostic categories. Each value represents the percentage of classification per object type across others. Rows indicate the ground-truth object class while columns display the object type predicted by the model. The diagonal values indicate the true positive portion for each object type and the other values, outside of the diagonal, display the misclassification rates. **c** Thumbnail images of 19 samples for cellular objects and non-cellular objects for cell detection and classification model.

During active learning, the trained YOLO model was applied on new tiles, especially those including the rare cellular and non-cellular objects to address class imbalances. The model's predictions were then converted to the readable format allowing hematopathologists and hematologists to evaluate the new output data and correct those objects missed or not classified correctly by the model (Fig. 4a). The new confirmed tiles were then merged with the current dataset to create a new (larger) dataset to train the model on this new dataset starting in the next iteration. The active learning cycle was performed until model performance plateaued. This approach resulted in increased training efficiency and model performance as measured by per day mAP, suggesting

**Table 3 Performance result of the proposed cell detection and classification model.**

| Object class | Precision | Recall | F1 score | Log-average miss rate | AP@0.5 |
|---|---|---|---|---|---|
| Neutrophil | 0.84 | 0.91 | 0.87 | 0.21 | 0.90 |
| Metamyelocyte | 0.68 | 0.79 | 0.73 | 0.37 | 0.77 |
| Myelocyte | 0.80 | 0.82 | 0.81 | 0.34 | 0.80 |
| Promyelocyte | 0.60 | 0.67 | 0.64 | 0.53 | 0.62 |
| Blast | 0.87 | 0.90 | 0.88 | 0.34 | 0.84 |
| Erythroblast | 0.86 | 0.92 | 0.89 | 0.17 | 0.92 |
| Megakaryocyte nucleus | 0.80 | 0.57 | 0.67 | 0.18 | 0.60 |
| Lymphocyte | 0.73 | 0.65 | 0.69 | 0.49 | 0.66 |
| Monocyte | 0.84 | 0.71 | 0.77 | 0.36 | 0.72 |
| Plasma cell | 0.75 | 0.69 | 0.72 | 0.33 | 0.72 |
| Eosinophil | 0.93 | 0.94 | 0.93 | 0.06 | 0.97 |
| Megakaryocyte | 1.00 | 0.79 | 0.88 | 0.19 | 0.82 |
| Debris | 0.85 | 0.80 | 0.82 | 0.34 | 0.79 |
| Histiocyte | 0.90 | 0.53 | 0.67 | 0.5 | 0.54 |
| Platelet | 0.84 | 0.64 | 0.73 | 0.33 | 0.64 |
| Platelet clump | 0.93 | 0.61 | 0.73 | 0.41 | 0.62 |
| **Average** | **0.83** | **0.75** | **0.78** | **0.32** | **mAP@0.5 =0.75** |

that an active learning approach could both augment training efficiency and model performance Table 4, Supplementary Table S4 and Fig. 4b). This process resulted in total of 1,178,408 objects annotated by the expert hematopathologists inside 132,000 ROI tiles (including augmentation).

**Summarizing bone marrow cytology as a histogram of cell types**. End-to-end model architecture for automated bone marrow cytology is shown in Fig. 1. After applying the end-to-end AI architecture (ROI detection and cell detection and classification models), a Histogram of Cell Types (HCT) is generated for each individual bone marrow aspirate ROI tile by counting all detected cellular and non-cellular objects in that tile. The individual ROI tile HCTs are then used to update an accumulated HCT, called Integrated Histogram of Cell Types (IHCT) summarizing the distribution of all cellular and non-cellular objects in all ROI tiles for a given patient, including the NDC (Fig. 5a). To assess for statistical convergence of individual HCTs to a final IHCT, after processing 80 tiles (≈800 cells), Chi-squared distance was calculated by adding each new ROI tile with an empirically determined threshold to assess when the IHCT is converged. Once it's converged, the bone marrow NDC is completed and is represented by the generated IHCT. Otherwise, another ROI tile is extracted and analyzed interactively until convergence. Based on analysis of 500 individual patients WSIs, we found that in most cases, IHCT convergence is reached after counting cells in 100–200 tiles for normal-diagnosed patients, 300–400 tiles in patients with a MDS-diagnosed patients and 400–500 tiles in patients with a AML-diagnosed patients (≈1000–5000 cells) (Fig. 5b). Looking for only a small number of tiles in this approach is time and computationally efficient in comparison to analyzing all regions across a WSI, similar to real-world clinical practice.

**Concordance between hematopathologists and model performance**. For comprehensive clinical evaluation of our end-to-end model, performance was evaluated by two additional hematopathologists who were not involved in cell labeling. Both hematopathologists showed high concordance with model performance, with mAP >90% overall cell and object types (Fig. 6 and and Supplementary Fig. S6 for details). Additionally, the Cohen's Kappa has been calculated both between the hematopathologists and model, and within the experts as well; 0.97 and

0.98 for expert 1 and expert 2, respectively and 0.99 between both experts.

**Discussion**

To date, limited studies have been performed toward automated bone marrow cytology, and despite the obvious clinical need there are currently no commercial computational pathology workflow support tools in this domain. This is likely due to the complex nature of aspirate specimens from a computer vision perspective compared to other cytology preparations such as peripheral blood specimens, where commercial support tools have existed for years [25]. In addition, adoption of digital pathology workflows has been slow. However, the field is showing increasing acceptance of digital pathology, which will enable a new generation of computational pathology workflow support tools [51,52]. Previous studies applying computational pathology to bone marrow aspirate cytology have focused only on cell classification versus automated end-to-end detection of ROI and cell types in aspirate specimens, which is essential for a viable workflow support tool. Choi et al. [53] proposed a method for cell classification in the NDC by applying dual-stage convolutional neural network (CNN). The dataset in their study comprised 2,174 cells from 10 cytological classes and did not include other important cellular and non-cellular object types in bone marrow cytology, such as histiocytes, megakaryocytes, and megakaryocyte nuclei which have high diagnostic relevance to hematology. Additionally, ROI tiles were detected manually by a human operator, abrogating utility as a clinical workflow support tool in hematology. Chandradevan et al. [26] developed a framework for bone marrow aspirate differential cell counts by using two-stage cell detection and classification deep learning models separately. However, this approach is operationally slow and only able to detect and classify 11 cell types, and again, entailed selection of ROI tiles manually by a human. Clinically relevant diagnostic workflow support tools for bone marrow aspirate cytology will need to fast (virtual real-time object detection and classification), accurate, and fully automated from end-to-end (i.e., from a raw digital WSI to analysis of bone marrow cell detection and classification).

Here, we present for the first time an end-to-end AI architecture for automated bone marrow cytology. This model performed well, with high accuracy and precision in both ROI detection and object classification in multiple clinical validation settings. Our model forms the basis for prototyping computational pathology clinical workflow support tools that support full automation in bone marrow cytology. This technology, when

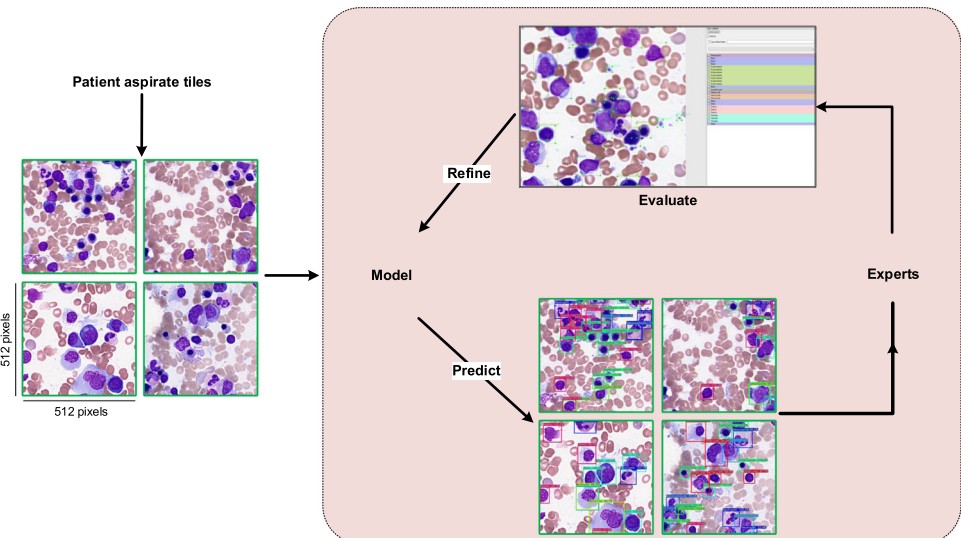

(a)

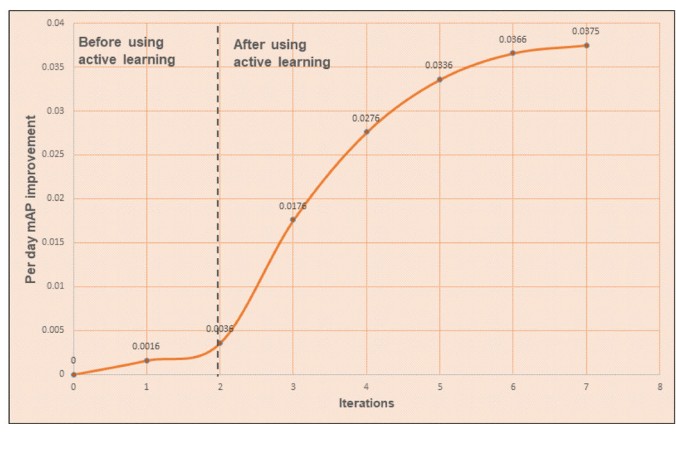

(b)

**Fig. 4 Model training started with a relatively small dataset and its performance increased by annotating more objects. a** A schematic of active learning process. **b** After using the active learning approach, per day mean average precision (mAP) has been improved drastically. Here in each iteration, 250 new tiles of mostly rare cellular and non-cellular objects are selected to be annotated and then merged to the current dataset. Without using active learning, all objects in the new tiles are annotated by the hematopathologists, but by using that, the model trained from the current dataset is run on the new tiles to detect and classify the objects and then hematopathologists review the results and confirm or modify them. In each iteration, the new annotated tiles are merged with the current dataset. The model will be trained on this new dataset and the next iteration will be started.

developed as commercial-grade workflow support tool may assist overworked pathologists both in busy reference centers, and less experienced pathologists in smaller, community centers. We additionally introduce several key advances into the field of AI-based computational pathology as applied to bone marrow cytology.

Firstly, we generate a novel information summary. The Histogram of Cell Types (HCT) representing the collective cytomorphological information in a patient bone marrow aspirate specimen. The Integrated Histogram of Cell Types (IHCT) represents a new framework for pathologists to interface with the complex information present in a cytology specimen, allowing for a rapid diagnostic assessment that can be integrated with other information (e.g., histomorphology) for augmented diagnostic

interpretation. Compared to a traditional NDC which consists of 300–500 manually counted cells, the proposed IHCT encompasses thousands of cells collected by statistical convergence. This augmented sensitivity may not only support more accurate and precise diagnosis in hematology, but also may eventually support the detection of rare cells that cannot be identified by human operators, such as "blasts" that constitute measurable residual disease (MRD) in acute leukemia.

Secondly, We use active learning to augment model performance. Our active learning approach allowed for rapid augmentation of model performance and training efficiency on a relatively small labeled dataset. The inability to annotate sufficient training data in specialized domains such as medicine is recognized as an impediment to generalizable and scalable deep

**Table 4 Performance result of using active learning.**

| Object class | Iteration 1 Count | AP | Iteration 2 Count | AP | Iteration 3 Count | AP | Iteration 4 Count | AP | Iteration 5 Count | AP | Iteration 6 Count | AP | Iteration 7 Count | AP | Iteration 8 Count | AP |
|---|---|---|---|---|---|---|---|---|---|---|---|---|---|---|---|---|
| Neutrophil | 680 | 0.75 | 1256 | 0.82 | 1568 | 0.83 | 1756 | 0.85 | 1895 | 0.86 | 2050 | 0.89 | 2398 | 0.91 | 2714 | 0.90 |
| Metamyelocyte | 480 | 0.60 | 605 | 0.66 | 752 | 0.69 | 785 | 0.72 | 856 | 0.76 | 925 | 0.75 | 986 | 0.76 | 1017 | 0.77 |
| Myelocyte | 390 | 0.53 | 589 | 0.55 | 665 | 0.59 | 720 | 0.62 | 869 | 0.70 | 950 | 0.78 | 1015 | 0.79 | 1199 | 0.80 |
| Promyelocyte | 65 | 0.44 | 102 | 0.46 | 256 | 0.52 | 285 | 0.54 | 320 | 0.59 | 326 | 0.62 | 360 | 0.64 | 409 | 0.62 |
| Blast | 1050 | 0.69 | 1785 | 0.76 | 2029 | 0.78 | 2590 | 0.81 | 2896 | 0.80 | 3268 | 0.83 | 3526 | 0.84 | 3950 | 0.84 |
| Erythroblast | 620 | 0.72 | 1150 | 0.78 | 1390 | 0.80 | 1580 | 0.82 | 2028 | 0.89 | 2295 | 0.90 | 2480 | 0.92 | 2668 | 0.92 |
| Megakaryocyte nucleus | 5 | 0.32 | 7 | 0.35 | 18 | 0.52 | 19 | 0.55 | 19 | 0.55 | 23 | 0.60 | 23 | 0.59 | 23 | 0.60 |
| Lymphocyte | 390 | 0.47 | 530 | 0.48 | 689 | 0.50 | 706 | 0.51 | 780 | 0.52 | 1015 | 0.59 | 1150 | 0.62 | 1305 | 0.66 |
| Monocyte | 62 | 0.47 | 98 | 0.51 | 295 | 0.57 | 368 | 0.61 | 423 | 0.62 | 485 | 0.65 | 520 | 0.68 | 569 | 0.72 |
| Plasma cell | 29 | 0.57 | 45 | 0.59 | 50 | 0.61 | 82 | 0.63 | 105 | 0.67 | 135 | 0.68 | 158 | 0.71 | 176 | 0.72 |
| Eosinophil | 31 | 0.59 | 38 | 0.63 | 135 | 0.83 | 172 | 0.86 | 185 | 0.88 | 221 | 0.95 | 228 | 0.95 | 249 | 0.97 |
| Megakaryocyte | 25 | 0.49 | 30 | 0.52 | 90 | 0.77 | 90 | 0.77 | 92 | 0.78 | 95 | 0.80 | 100 | 0.81 | 106 | 0.82 |
| Debris | 1380 | 0.58 | 2680 | 0.62 | 3450 | 0.65 | 3920 | 0.68 | 4490 | 0.73 | 4901 | 0.77 | 5260 | 0.77 | 5603 | 0.79 |
| Histiocyte | 38 | 0.34 | 72 | 0.42 | 147 | 0.48 | 163 | 0.48 | 168 | 0.51 | 174 | 0.52 | 182 | 0.54 | 191 | 0.54 |
| Platelet | 790 | 0.41 | 1680 | 0.46 | 2150 | 0.48 | 2560 | 0.52 | 2890 | 0.58 | 3250 | 0.65 | 3680 | 0.65 | 3971 | 0.64 |
| Platelet clump | 93 | 0.37 | 146 | 0.41 | 320 | 0.54 | 409 | 0.56 | 475 | 0.57 | 536 | 0.58 | 563 | 0.61 | 585 | 0.62 |
| **Average** | **6128** | **0.52** | **10813** | **0.56** | **14004** | **0.64** | **16205** | **0.66** | **18491** | **0.69** | **20649** | **0.72** | **22629** | **0.74** | **24735** | **0.75** |

Model training started with a small dataset at the first and second iteration, and then is improved (especially on rare cellular objects) in the subsequent iterations by using active learning.

learning approaches. Our approach suggests computational pathology workflow support tools could be designed from a human-centric AI perspective, where expert pathologists continuously evaluate and improve model performance in the context of a clinical diagnostic workflow.

Thirdly, we lay the basis for a commercial-grade diagnostic workflow support tool in hematopathology that may for example, be integrated with a hardware product such as digital scanner to acquire and analyze only the ROI relevant for diagnosis. This would not only speed up diagnostic workflows, where even in compressed form, aspirate specimen WSI range from 5-10 GB in size, but also has implications on efficient data storage and retrieval, which may be impediments to adoption of digital workflows in cytology[54,55].

Potential weaknesses include overfitting of our model to our local dataset, which is noted problem in AI-based computational pathology studies. As well, annotated publicly available or even academic digital pathology datasets are not yet widely available. Given the rarity of digital hematopathology workflows, particularly in aspirate cytology, external validation was not feasible at this early stage, however, will be essential in the development of a clinical-grade prototype. We expect this to improve as digital pathology workflows are increasingly adopted, supporting collaborative and robust validation of computational pathology tools. Additionally, some cell and object types, such as megakaryocyte nucleus and histiocyte, performed with moderate to low mAP. This is likely due to the rarity of these objects, and performance may improve with access to large training datasets. Specifically, blasts and lymphocytes showed overlap in classification and lower accuracy by our model (Fig. S2, which is a similar problem to human hematopathologists. A proportion of Plasma cells were also misclassified as erythroblasts in cases diagnosed as plasma cell neoplasm (Fig. S2). This may reflect biases in model performance, or alternatively, may be a function of the overlapping cytological features in these cell types which are often confused in clinical practice, specifically in MDS where dysplasia renders morphology challenging. We acknowledge this as a weakness in our model, one that is somewhat mitigated in real-world clinical practice by expert human hematopathologists using integrated, semantic interpretation of multiple ancillary data modalities, such as flow cytometry, molecular studies and clinical findings. As an

early prototype, this problem may be addressed in future work by incorporating additional training data, as well as multi-model ML-analyzed datasets and active learning approaches. It is critical to emphasize our ML technology would be intended to support pathologists and expedite workflows, requiring substantive human oversight and rigorous clinical validation, especially where blast counts represent critical diagnostic cutoffs in diagnosing MDS and acute leukemias.

Morphology in MDS poses a challenge for even the most experienced pathologists, with high inter-observer variability, as reflected in our model performance of MDS cases (Fig. S2b). This is a potential problem in a model that uses discrete class probability assignments for individual cells, where there still may be significant intra-class heterogeneity. Future iterations of our model may use approaches such as deep feature extraction from YOLO and dimensionality reduction to explore unsupervised relationships between cells in each class, ex, dysplasia within neutrophils, which may assist pathologists in interpreting cell subsets in cases with morphological dysplasia. This type of approach would yield additional information that goes beyond simple class probability assignment, allowing pathologists to understand and visualize cytological relationships learned by a model. One could envision a multi-modal deep learning approach to hematopathology workflows, integrating rich, deep information from multiple data sources, including histopathology (the trephine core biopsy), flow cytometry, molecular and clinical data to provide an overall semantic-level, and attentive diagnostic prediction and interpretation. Furthermore, the extracted multi-omics information would have high predictive and prognostic potential when linked to clinical outcomes and pharmacological responses.

The following software tools have been utilized at this work: Tensorflow 2.2.0 framework, OpenCV 4.1 library, C++, OpenSlide 3.4.1 and Large-Image 1.5. Regarding hardware, a computer server with Xeon CPUs, 4 GPUs Tesla V100 32 GB, and 128 GB RAM have been used to train the entire architecture, and for model deployment and production phase, a notebook with Intel Core i9 processor, 64 GB RAM and NVIDIA Quadro RTX 4000 8GB has been used. Regarding the execution time in model deployment and production phase, it took approximately 4min to generate the appropriate tiles (includes reading a digital WSI

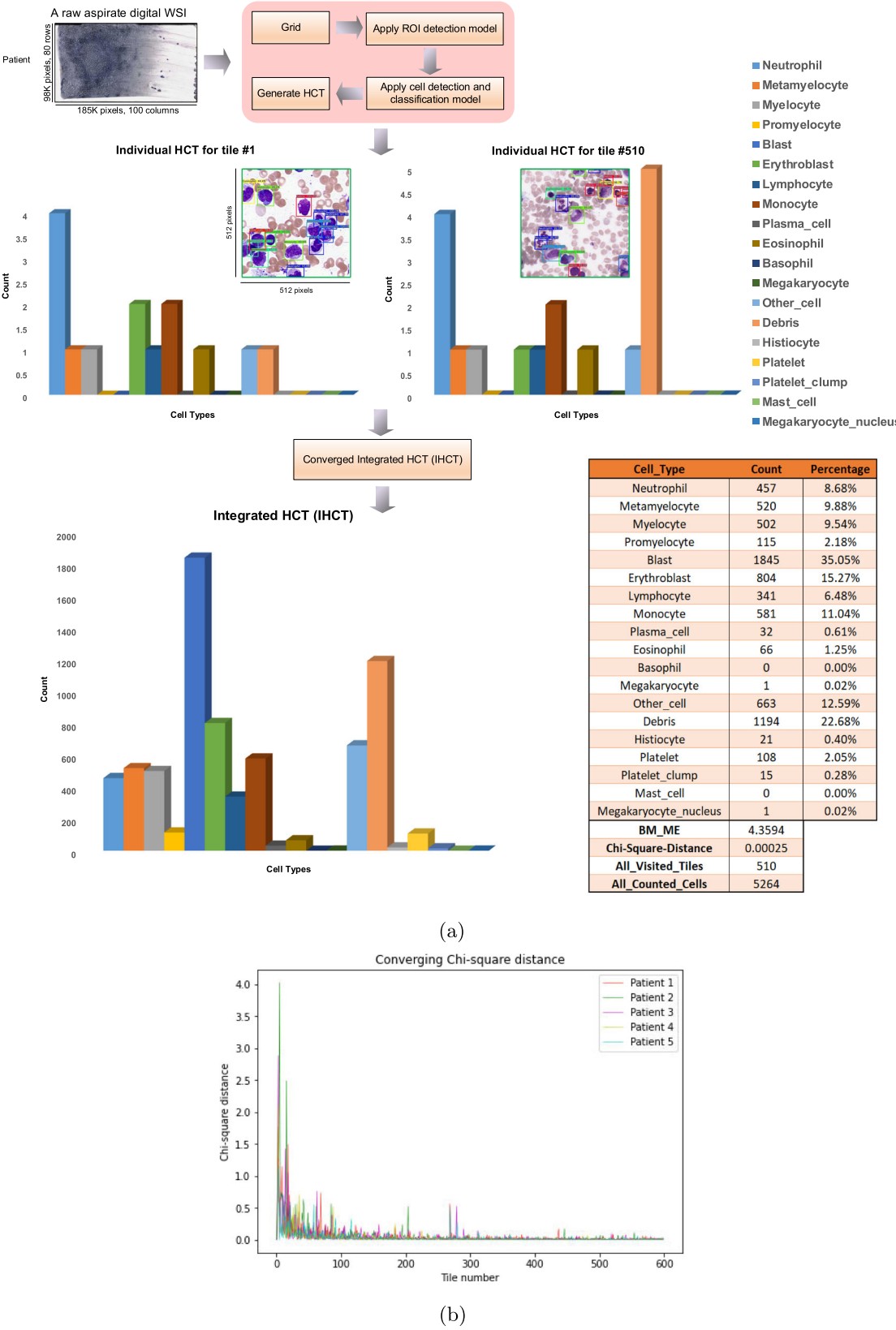

**Fig. 5 Generating the Histogram of Cell Types (HCT) and converged Integrated Histogram of Cell Types. a** Each histogram of cell types (HCT) is created individually for each Region of Interest (ROI) tile. The Integrated HCT (IHCT) is then updated as successive ROI histogram are accumulated. This process is stopped once the IHCT is converged using the Chi-square ($\chi^2$) distance. Collective cytological information from a patient bone marrow aspirate is then represented as an IHCT and a table with summary statistics such as the number of each cell type, percentage, BM$_{ME}$ ratio and Chi-Square distance. **b** Variation of the Chi-square distance through visiting and analyzing selected ROI tiles by the architecture and how an IHCT is converged by processing each tile. For five samples of patients, the IHCT is converged after visiting a number of tiles in the range of 400 to 500 tiles.

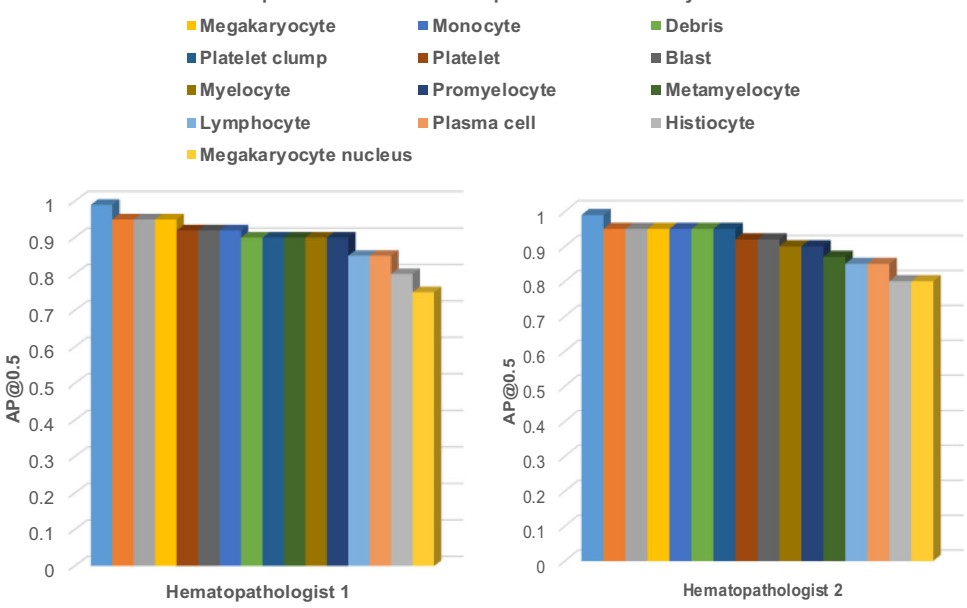

**Fig. 6 Concordance between hematopathologists evaluation and the obtained results.** Hematopathologists' evaluation for the cell detection and classification based on calculated average precision (AP) for each class.

from disk, creating the tiles with a size of 512*512 pixels, and applying the proposed ROI detection model). Consequently, the whole process to examine each tile takes about 30 milliseconds. For cell detection and classification, 50 milliseconds on average take for both detection and classification in each tile. As in most cases, the IHCT is converged in almost 400 to 500 tiles, the whole process of generating the IHCT took approximately 5min.

## Data availability
The data that support the findings of this study are available on reasonable request from the corresponding author, pending local REB approval. The data are not publicly available due to them containing information that could compromise research participant privacy/consent. Source data underlying the main figures in the manuscript are available as 'Supplementary Information'.

## Code availability
Code for this study is available at https://doi.org/10.5281/zenodo.6373429[56].

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

## Acknowledgements
This work was supported by a New Frontiers in Research Fund Exploration Grant (NFRFE), an Ontario Molecular Pathology Research Network (OMPRN) Cancer Pathology Translational Research Grant (CPTRG), and a Canadian Cancer Society BC Sparks Grant.

## Author contributions
R.M.T. designed and implemented the computational workflow, contributed to algorithm selection, designed and conducted experiments, analyzed data and contributed to writing the manuscript; Y.M. and T.D. provided critical review of the manuscript; C.R., M.S. and R.F. analyzed model performance; H.R.T. conceived the initial ideas and the overall approach, oversaw technical aspects of the project, designed experiments, and contributed to writing the manuscript; C.J.V.C. conceived the initial ideas and the overall approach, oversaw medical aspects of the project, analyzed and annotated data, designed experiments, and contributed to writing the manuscript.

## Competing interests
The authors declare no competing interests.
