## [Peer Review File · Communications Medicine]

Reviewers' comments:

Reviewer #1 (Remarks to the Author):

This study tries to construct an automatic end-to-end solution to analyze the bone marrow smear, which is important for clinical practice. Currently, there is no satisfactory product to conduct this work, and it is really labor intensive. However, the number of leukemia cases is related small in clinical practice, and this product may only save a small part of cost for a whole hospital. Moreover, there are some previous researches to address the same issue. Therefore, the scientific significance of this study is relatively low. Of course, author demonstrated a complete experiment including high level techniques and the-state-of-the-art method. In summary, the novelty of this study is low but the experimental technology is acceptable. The results were also reliable if author can provide more details as follows.

Major comments

1. The term "Histogram of Cell Types" is misleading due to a specific term of "Image histogram" in computer vision domain. In fact, it just a distribution of each cell type. Please revise the name of your paper.
2. I think the data splitting needs to be clarified. As far as I understand, data was spited into 5 subsets for 5-fold cross-validation at random. If that is correct, how do you address data leakage, i.e., how do you make sure that your model does not use information from the same patient? Is there any patient that is included in both, the training and test set?
3. This study only included one dataset. The experiment should be included a training set for backpropagation, a validation set for hyper-parameter selection, and a test set for performance assessment. I understand this study is try to use 5-fold cross validation to sidestep this issue. However, no details about hyper-parameter selection let me suspect to overestimate the model performance. Author should re-conducted the experiments to follow the standard process.
4. There were 3 iterations for data collection in this study. Please provided the performance details of each iteration. For example, how many samples (especially in the sample of rare cell type) were collected from first, second, and third iterations for training and validation, and the performance should be provided. Importantly, is the validation set vary in each iteration?
5. YOLO allows thresholds on the class probability scores for its bounding box objects. Were these thresholds explored (e.g. require a probability of > 0.80 to classify as a particular cell type, otherwise unable to identify), or was the class with the highest probability simply chosen? Were there any other notable settings/parameters used for YOLO here?
6. This study used 400x image to conduct experiments. However, certain cell type identification required 1000x image for experienced hematologists. How to make sure that experts correctly identify all cell types?
7. In fact, the result of flow cytometry is the only gold standard to describe the distribution of cell types. I considered the comparison between AI model and hematologists cannot provide the enough evidence to explain the accuracy. This is an important limitation in this study. A viable alternative is to conduct an expensive human-machine competition including more than 5 hematologists, especially in physicians who have not participated in the training stage.

Minor comments

8. Authors described "However, to date, YOLO has not been applied to medical domain problems

such as pathology." in introduction section. However, I can easily find related paper in PubMed [JMIR Med Inform. 2020 Apr 8;8(4):e15963. doi: 10.2196/15963.]. Please conduct a complete review and discuss the novelty of this study compared to previous researches.

9. The experiment details to make the bone marrow smear is important to repeat the works. However, there is not enough details, especially in the bone marrow aspiration, staining, and digitalized.

10. The definition of the YOLO bounding box should be provided, in particular the settings of anchor boxes. The authors may wish to clean up them.

11. The authors might provide examples of misclassified cells, and discuss whether the misclassifications are egregious, or reasonable for human experts.

Reviewer #2 (Remarks to the Author):

Results from the patient-level cross validation need to be shown. Please also perform a leave-one patient-out cross-validation and add results to the manuscript. This comment pertains both to ROI, non-ROI classification results and to the cell-based classifications as well.

How are the classification metrics generated? Does for instance the reported F1-score mean an average F1score for a single cell type with the average calculated over all regions and WSIs? Same question pertains to the ROI vs. non-ROI classification metrics.

What is mAP@0.5? Non computer science audience may not be familiar with this term. Please be clearer with the methods description.

It is unclear how the IHCT convergence is accomplished. It looks like this is an iterative process but initial conditions of the convergence assessment are not described. How many regions with their respective HCTs are required to begin with? How many ROIs are required to achieve convergence? What is the minimal and maximal number of ROIs to assure that convergence can be achieved? The authors provided ranges of ROI and cell types for the IHCT convergence to be achieved, but it is unclear whether these numbers can be universally used (or observed) for aspirates from normal vs. abnormal (i.e containing abnormal cells) bone marrow. Why measuring convergence is even needed here? BTW, why a simple histogram of cells from all regions across a WSI is not a good way of representing the cell frequencies, and why IHCT would be better than this histogram?

What is the value of MSE measured between NDCs by expert 1 and 2 ? Can the authors provide a quantitative evidence that the human operator-performed NDC is subject to increased intra-category variability?

I doubt section 2.5 can be called clinical validation of model performance. What the authors did here is the identification of differences between human counts vs. IHCT yielded by AI by MSE. A clinical validation would include statistical tests showing that results of clinical decisions made based on IHCT are similar/concordant or same when compared to those based on manual NDC. At this point, it is also unclear how useful IHCT would be comparing to NDCs clinically.

The authors said that they used 100 samples (1 sample per patient) in their experiment. What kind of samples were used here (say how many from normal bone marrows vs., abnormal ones) ? What MESs per group would be like when this potentially non-uniform sample of aspirates is broken down

into clinically meaningful groups?

Another problem here is that the authors used manual ground truth to train and validate their AI. However, manual cell labeling is inaccurate and has high intra and inter-rater variability. A more appropriate would be to use immunohistochemistry to label cells for training, validation and testing purposes. An example of such approach is shown here:

<https://www.ncbi.nlm.nih.gov/pmc/articles/PMC7177428/>

Since immunohistochemistry (IHC) is likely to be more accurate for cell labeling, the authors should show classification results by their AI on IHC labeled cells or aspirates as the ground truth.

Active learning is not necessarily a new approach in digital pathology. It has been or recommended for use before. See these example papers: https://link.springer.com/chapter/10.1007/978-3-030-23937-4_3

<https://onlinelibrary.wiley.com/doi/full/10.1002/path.5331>

<https://pubmed.ncbi.nlm.nih.gov/32758706/>

Inaccurate or unproven statements: (1) Abstract: "HCT has potential to revolutionize hematopathology diagnostic workflows, leading to more cost-effective, accurate diagnosis and opening the door to precision medicine". (2) Pg.2: "YOLO has not been applied to medical domain problems such as pathology". Here are examples where YOLO was used in digital pathology.

<https://pubmed.ncbi.nlm.nih.gov/31476576/>,

<https://www.osapublishing.org/osac/fulltext.cfm?uri=osac-4-2-323&id=446861> and there are more.

Reviewer #3 (Remarks to the Author):

The authors create a tool for automated evaluation of bone marrow aspirate smears using deep learning methodologies. In their method, they identify appropriate regions of interest and use the YOLO model to classify cells in the smears. Their method was improved through active learning through iterative review of model output by hematopathologists. They ultimately validated their model through the review of 2 additional hematopathologists reviewing 100 patient samples.

Overall, this paper describes an exciting new tool for automated bone marrow aspirate differential that represents a great opportunity for AI to assist in and streamline diagnoses from bone marrow samples. The authors have dealt with one of the major challenges of bone marrow aspirate evaluation, the identification of appropriate regions of interest, and have demonstrated the benefits of active learning in improving performance of the model.

Critiques:

- 1) The authors do not address either the types of disease entities used in this study or how different morphologic abnormalities might affect its performance. The numbers of each disease reviewed are not described. For example, how many smears represented myelodysplastic syndrome (MDS) and did abnormal morphology seen in MDS affect precision? I would expect it to. Furthermore, how many pathologist/AI discrepancies resulted from low numbers of a cell type (as mentioned in the discussion). E.g., might some of the discrepancy reflect enhanced diagnostic ability through counting of so many cells (and detection of very rare blasts not seen by the human, for example)?
- 2) How did the model handle other/unidentifiable cells (e.g. cells with very abnormal morphology).
- 3) The authors raise the issue of computational time in the introduction. They should describe the

run time and whether or not it is feasible to implement in clinical practice.

4) There is a caveat in the discussion that is difficult to follow: "Finally, when compared to manual NDC in 100 patients, the model performed well..." This part seems to suggest the possibility of interobserver variability; though a high degree of interobserver variability would be unexpected in the identification of neutrophils in particular. It would be best for the authors to write these two sentences more clearly as they are hard to follow. But also, is it possible that because neutrophils are so frequent in aspirate smears, that it might affect the variance in counts, contributing to the observations described (or maybe not, just a thought).

All major changes in the revised manuscript have been **highlighted in yellow**.

Reviewer #1 (Remarks to the Author):

This study tries to construct an automatic end-to-end solution to analyze the bone marrow smear, which is important for clinical practice. Currently, there is no satisfactory product to conduct this work, and it is really labor intensive. However, the number of leukemia cases is related small in clinical practice, and this product may only save a small part of cost for a whole hospital. Moreover, there are some previous researches to address the same issue. Therefore, the scientific significance of this study is relatively low. Of course, author demonstrated a complete experiment including high level techniques and the-state-of-the-art method. In summary, the novelty of this study is low but the experimental technology is acceptable. The results were also reliable if author can provide more details as follows.

We thank the reviewer for these comments. As mentioned in the introduction, the scope of this work goes beyond only blast counts in leukemias, which are only a small proportion of clinical hematopathology practice. In the US, there are approximately 700 000 bone marrow studies performed annually, and only a fraction of these are leukemias. There is no comparable estimate for Canada, but at 1/10 the population this would be in the range of 70 000. This amounts to approximately 800 000 bone marrow studies performed annually in North America, each one requiring aspirate cytology review. Therefore, we foresee significant cost savings as hematopathologists would be able to take on increased caseload with clinically validated ML-based cytology tools. To our knowledge, reliable ML technology to perform aspirate NDC in an actual clinical setting has not yet been validated, or even demonstrated. We hope this helps to clarify the potential **novelty and clinical significance** of this work. **We now mention this in the introduction, page 1 of the revised manuscript.**

Major comments

1. The term "Histogram of Cell Types" is misleading due to a specific term of "Image histogram" in computer vision domain. In fact, it just a distribution of each cell type. Please revise the name of your paper.

Thank you for the comment. We feel that for the purposes of this work, the terms distribution and histogram are overlapping and analogous terms. As this paper is clinically-oriented, rather than oriented at the ML community, the term histogram conveys the purpose of the work clearly to the clinical/pathologist audience, i.e., we are automating a bone marrow manual nucleated differential count which is a distribution of cells in various categories as a histogram. Similarly, a manual nucleated differential count can be called a histogram in clinical medicine. We therefore feel that "Histogram of Cell Types" is the most accurate and meaningful way to convey our technology and results to our clinical colleagues. This should not be confusing to our readers, as the primary audience in this journal will be clinical pathologists versus deep learning / computer vision specialists. As well, it is generally understood that a "histogram" is a quantification of counting discretized measurements. We also added in Introduction: **"A histogram is generally a representation of a distribution, a very old graphical technique to count discrete values [37]."**

2. I think the data splitting needs to be clarified. As far as I understand, data was spited into 5 subsets for 5-fold cross-validation at random. If that is correct, how do you address data leakage, i.e., how do you make sure that your model does not use information from the same patient? Is there any patient that is included in both, the training and test set?

Thank you for the helpful comment. In this work, the dataset has been split into train, validation and test sets at patient level. It means each set has a unique patient WSI that does not occur in the other sets. This strategy has been applied on both “ROI detection” and “cell detection and classification” models, such that there is not any data leakage between the sets. We updated the text and highlighted it in the manuscript, at sections 4.3 and 4.4, as follows:

In section 4.3 page 16:

“We applied patient-level 5-folds cross-validation to train and test the model. Hence, the dataset (98,750 tiles) was split into two main partitions in each fold, training and test-validation, 80% (204 WSIs, including 80,250 tiles) and 20% (46 WSIs including 18,500 tiles), respectively. The test-validation was also been split into two main partitions, 70% validation and 30% test. To ensure that enough data for each class was chosen in our dataset, the above split ratios were enforced on appropriate and inappropriate tiles separately. The dataset was split into training, validation and test sets at patient level, such that each set has a patient WSI that does not come in the other sets to prevent data leakage. In each fold, the best model was picked by running on the validation partition after the training and then evaluated on unseen patients in the test dataset.”

In section 4.4 page 16:

“Similar to the ROI detection method above, patient-level 5-folds cross-validation was applied to train the model here. Therefore, each fold is divided into training and test-validation partitions, 80% and 20% respectively. The test-validation data portion was split into two main partitions (70% validation and 30% test). Additionally, to ensure that enough data for each class was chosen in our dataset, the mentioned portions were enforced on each object class type individually. In each fold, the best model was picked by running it on the validation partition and then evaluation on the test (unseen) dataset was performed using the mean average precision (mAP).”

3. This study only included one dataset. The experiment should be included a training set for backpropagation, a validation set for hyper-parameter selection, and a test set for performance assessment. I understand this study is try to use 5-fold cross validation to sidestep this issue. However, no details about hyper-parameter selection let me suspect to overestimate the model performance. Author should re-conducted the experiments to follow the standard process.

Thank you for the comment. The dataset of 250 patient WSIs randomly selected from over 1000 WSI were divided into Train, Validation and Test sets for the ROI detection model, and 106 randomly selected patient WSIs were divided into Train, Validation and Test sets for YOLO model development. We highlighted this (as in the previous comment) in sections 4.3 and 4.4 of the manuscript. Also, regarding hyper-parameters, we highlighted it in the manuscript with the following sentences:

In section 4.3 page 15:

“This network was trained using a cross entropy loss function and AdamW optimizer with learning rate $1e-4$ and weight decay $5.0e-4$. Also, a pretrained DenseNet121 was applied to initialize all weights in the network prior to fine-tuning. The entire network was fine-tuned for 20 epochs with 32 batch size.”

In section 4.4 page 16:

“In addition, the hyperparameters for bone marrow cell detection and classification were used as follows: max-batches is 130,000; the training steps are 104,000 and 117,000; batch size 64 with subdivision 16; the polynomial decay learning rate scheduling strategy is applied with an initial learning rate of 0:001; the momentum and weight decay are set as 0:949 and 0:0005 respectively; warmup step is 1,000; YOLO network size set to 512 in both height and width; anchor size set to 13, 14, 19, 18, 29, 30, 19, 64, 62, 20,

41, 39, 35, 59, 50, 49, 74, 35, 56, 62, 68, 53, 46, 87, 70, 70, 95, 65, 79, 85, 101, 95, 87, 129, 139, 121, 216, 223.”

4. There were 3 iterations for data collection in this study. Please provided the performance details of each iteration. For example, how many samples (especially in the sample of rare cell type) were collected from first, second, and third iterations for training and validation, and the performance should be provided. Importantly, is the validation set vary in each iteration?

Thank you very much for the comment. The details of each iteration have been provided inside the manuscript in **Tables 3 page 4**. Regarding the second question, yes, in each iteration a new Training, and Test-Validation sets were created and the model trained and evaluated on them. Regarding “samples of rare cell types”; we randomly selected patient WSI samples from our starting dataset. We assume by rare cell types, you mean cells such as blasts, megakaryocytes or histiocytes. As can be seen in Table 3, these were all sampled proportionally across iterations.

5. YOLO allows thresholds on the class probability scores for its bounding box objects. Were these thresholds explored (e.g. require a probability of > 0.80 to classify as a particular cell type, otherwise unable to identify), or was the class with the highest probability simply chosen? Were there any other notable settings/parameters used for YOLO here?

Thank you for the comment. The following sentence has been added and highlighted in the manuscript (**section 4.5 page 17**):

“It is worth mentioning that the value of 0.5 was considered for Intersection over Union (IoU) in AP for each object detection and >0.75 has been used for class probability.”

Also other YOLO parameters have been described as follows, at section 4-4 of the manuscript at page 16:

“In this architecture, CSPDarknet53 [46] was used as the backbone of the network to extract features, SPP [47] and PAN [48] were used as the neck of the network to enhance feature expressiveness and robustness, and YOLOv3 [49] as the head. As bag of specials (BOS) for the backbone, Mish activation function [50], cross-stage partial connection (CSP) and multi input weighted residual connection (MiWRC) were used. For the detector, Mish activation function, SPP-block, SAM-block, PAN path-aggregation block, and DIoU-NMS [51] were used. As bag of freebies (BoF) for the backbone, CutMix and Mosaic data augmentations, DropBlock regularization [52], and class label smoothing were used. For the detector, complete IoU loss (CloU-loss) [51], cross mini-Batch Normalization (CmBN), DropBlock regularization, Mosaic data augmentation, self-adversarial training, eliminate grid sensitivity, using multiple anchors for single ground truth, Cosine annealing scheduler [53], optimal hyperparameters and random training shapes were used .”

6. This study used 400x image to conduct experiments. However, certain cell type identification required 1000x image for experienced hematologists. How to make sure that experts correctly identify all cell types?

Thank you for the comment. The 4 Royal College of Physicians and Surgeons of Canada-certified clinical hematopathology and hematology members of our team can attest that in modern hematopathology practice for bone marrow cytology **1000X oil-immersion is not required in routine practice**. Furthermore, no digital scanning hardware currently exists that we are aware of that captures 1000X images for cytology. We therefore feel using 400X image magnification accurately reflects current best practice in diagnostic hematopathology. The expertise for identification comes from many years of training and board certification, the same standard that is used for actual clinical diagnosis.

7. In fact, the result of flow cytometry is the only gold standard to describe the distribution of cell types. I considered the comparison between AI model and hematologists cannot provide the enough evidence to explain the accuracy. This is an important limitation in this study. A viable alternative is to conduct an expensive human-machine competition including more than 5 hematologists, especially in physicians who have not participated in the training stage.

Thank you for the comment. With due respect, to clarify from the perspective and knowledge of the numerous hematopathologists and hematologists involved in this work, **flow cytometry is not the reference standard for cell distributions in clinical diagnostic hematopathology.** The reference standard for bone marrow cell distributions is expert-pathologist cell counts and annotations. This is documented in ICSH International Guidelines (1.LEE, S. -H., ERBER, W. N., PORWIT, A., TOMONAGA, M. & PETERSON, L. C. ICSH guidelines for the standardization of bone marrow specimens and reports. *Int J Lab Hematol* **30**, 349-364 (2008). Therefore, pathologist annotation for pathology images is considered the ground truth; this is well-established in the field, and it is the benchmark for clinical diagnosis and patient care. In fact, the flow cytometry specimen often contains hemodilution due to being collected in last order during a bone marrow exam, and therefore is notoriously inaccurate regarding cell counts. This may be different from the research domain. We therefore feel that the cell annotation and annotation evaluation of our model by our team of certified expert hematopathologists and hematologists with between 5-35 years of morphology experience is sufficient as a reference standard for our work. This again would agree with international best-practice clinical guidelines. **We clarify the rationale for expert-pathologist cell annotation in the methods section, section 4.2 page 14, now entitled “Data Annotation and Augmentation Strategy.”**

The idea of a competition study, while conceptually good, would not be possible in the scope of this work and as the reviewer points out, would be prohibitively expensive and likely unnecessary. We also had 2 hematopathologists who did not participate in initial annotation evaluate model performance. **We describe it as follows in section 2.5 at page 12 of the manuscript.**

“For comprehensive clinical evaluation of our end-to-end model, performance was evaluated by two additional hematopathologists who were not involved in cell labeling. Both hematopathologists showed high concordance with model performance, with mAP > 90% overall cell and object types (Fig. 6 for details). Additionally, the Cohen’s Kappa has been calculated both between the hematopathologists and model, and within experts as well; 0.97 and 0.98 for expert 1 and expert 2, respectively and 0.99 between both experts.”

Minor comments

8. Authors described "However, to date, YOLO has not been applied to medical domain problems such as pathology." in introduction section. However, I can easily find related paper in PubMed [JMIR Med Inform. 2020 Apr 8;8(4):e15963. doi: 10.2196/15963.]. Please conduct a complete review and discuss the novelty of this study compared to previous researches.

Thank you for the great comment. The manuscript has been modified by adding the following sentence in **section 1 at page 2:**

“For example in [36], YOLO has been applied to assess the cell types in blood smears. However, only 7 cell types have been considered in that study. Moreover, the tiles need to be selected manually by the user.

9. The experiment details to make the bone marrow smear is important to repeat the works. However, there is no enough details, especially in the bone marrow aspiration, staining, and digitalized.

Thank you very much for the comment. The details of making a bone marrow smear is critically important, we agree. However, detailing a well-accepted and standardized process is not within the scope of the paper. The technical details of making a bone marrow smear can be found at the following references:

[1] LEE, S. -H., ERBER, W. N., PORWIT, A., TOMONAGA, M. & PETERSON, L. C. ICSH guidelines for the standardization of bone marrow specimens and reports. *Int J Lab Hematol* **30**, 349-364 (2008).

[4] Bain, B. Bone marrow aspiration. *Journal of clinical pathology* **54**, 657-663 (2001).

This is described and referenced in **section 2.1 page 3 of the revised manuscript**.

10. The definition of the YOLO bounding box should be provided, in particular the settings of anchor boxes. The authors may wish to clean up them.

Thank you for the comment. It's added in the manuscript and highlighted, as follows:

In section 4.4 page 16:

“anchor size set to 13, 14, 19, 18, 29, 30, 19, 64, 62, 20, 41, 39, 35, 59, 50, 49, 74, 35, 56, 62, 68, 53, 46, 87, 70, 70, 95, 65, 79, 85, 101, 95, 87, 129, 139, 121, 216, 223

11. The authors might provide examples of misclassified cells, and discuss whether the misclassifications are egregious, or reasonable for human experts.

Thank you for the comment. We address it as a confusion matrix in **Figure 2(b), at page 6, and also Figure S2 at page 23**. Also in section 3 (page 13), we discussed this specifically issue in the text, as follows **on page 13 of the revised manuscript**:

“Interestingly, “blasts” and “lymphocytes” showed some overlap in classification by our model, which is a similar problem to human hematopathologists. This may reflect biases in model performance, or alternatively, may be a function of the overlapping cytological features in these cell types which are often confused in clinical practice

Reviewer #2 (Remarks to the Author):

1) Results from the patient-level cross validation need to be shown. Please also perform a leave-one patient-out cross-validation and add results to the manuscript. This comment pertains both to ROI, non-ROI classification results and to the cell-based classifications as well.

Thank you very much for the comment. The results for both “ROI detection” and “cell detection and classification” models are patient-level cross validation, such that the dataset was split into train, validation and test sets at patient level. It means each set has patients that do not occur in the other ones. This strategy was applied on both “ROI detection” and “cell detection and classification” models. Therefore, this approach includes “leave one patient out”. **To clarify this, we updated the text and highlighted it in the manuscript, at sections 4.3 and 4.4, as follows:**

In section 4.3 page 16:

“We applied patient-level 5-folds cross-validation to train and test the model. Hence, the dataset (98,750 tiles) was split into two main partitions in each fold, training and test-validation, 80% (204 WSIs, including 80,250 tiles) and 20% (46 WSIs including 18,500 tiles), respectively. The test-validation also has been split into two main partitions, 70% validation and 30% test. To ensure that enough data for each class has been chosen in our dataset, the above split ratios were enforced on appropriate and inappropriate tiles separately. It is worth mentioning that the dataset has been split into training, validation and test sets at patient level, such that each set has a patient that does not come in the other sets to prevent data leakage. In each fold, the best model was picked by running on the validation partition after the training and then evaluated on unseen patients in the test dataset”

In section 4.4 page 16:

“Similar to the ROI detection method above, patient-level 5-folds cross-validation was applied to train the model here. Therefore, each fold is divided into training and test-validation partitions, 80% and 20% respectively. The test-validation data portion was split into two main partitions (70% validation and 30% test). Additionally, to ensure that enough data for each class was chosen in our dataset, the mentioned portions were enforced on each object class type individually. In each fold, the best model was picked by running it on the validation partition and then evaluation on the test (unseen) dataset was performed using the mean average precision (mAP).”

2) How are the classification metrics generated? Does for instance the reported F1-score mean an average F1 score for a single cell type with the average calculated over all regions and WSIs? Same question pertains to the ROI vs. non-ROI classification metrics.

This is an excellent question, and perhaps we did not clarify this in the paper. All metrics were calculated for each single cell type separately and then average values for all these are subsequently calculated. For ROI vs. Non-ROI, as it is only two classes, so all the metrics already calculated only on average. We updated the text and highlighted in the manuscript as follows:

In abstract at page 1:

“The approach achieves high accuracy in region detection (0.97 accuracy and 0.99 ROC AUC), and cell detection and cell classification (0.75 mAP, 0.78 average F1-score, Log-average miss rate of 0.31).”

Also in the last paragraph in Introduction, at page 3:

“Our approach shows cross-validation accuracy of 0.97 and precision of 0.90 in ROI detection (selecting appropriate tiles), and mAP of 0.75 and average F1-score of 0.78 for detecting and classifying 16 key cellular and non-cellular objects in aspirate WSIs.”

In section 2.2 at page 5:

“The model achieved a high mean Average Precision (mAP) and average F1 score in object detection and classification: mAP, average F1-score, precision and recall are 0.75, 0.78, 0.83, and 0.75, respectively”

3) What is mAP@0.5? Non computer science audience may not be familiar with this term. Please be clearer with the methods description.

Thank you for the comment. We explained mAP and other metrics in the manuscript. **It is highlighted in section 4.5 (Evaluation) at page 17 and 18 of the revised manuscript.**

“To assess the performance of the proposed cell detection and classification method, Average Precision (AP) was used with 11-point interpolation (Eq. 8). Also at the end, the mean Average Precision (mAP) [54] was calculated for all the AP values (Eq. 10). The value of recall was divided from 0 to 1.0 points and the average of maximum precision value was calculated for these 11 values. It is worth mentioning that the value of 0.5 was considered for Intersection over Union (IoU) in AP for each object detection and >0.75 has been used for class probability. In addition, Precision, Recall, F1-score (Eq. 12), average IoU (Eq. 11) and log-average miss rate (Eq. 13) have been calculated here for each object type.”

4) It is unclear how the IHCT convergence is accomplished. It looks like this is an iterative process but initial conditions of the convergence assessment are not described. How many regions with their respective HCTs are required to begin with? How many ROIs are required to achieve convergence? What is the minimal and maximal number of ROIs to assure that convergence can be achieved? The authors provided ranges of ROI and cell types for the IHCT convergence to be achieved, but it is unclear whether these numbers can be universally used (or observed) for aspirates from normal vs. abnormal (i.e containing abnormal cells) bone marrow. Why measuring convergence is even needed here? BTW, why a simple histogram of cells from all regions across a WSI is not a good way of representing the cell frequencies, and why IHCT would be better than this histogram?

Thank you very much to the reviewer for bringing up these important points. We address the points raised as follows:

- It looks like this is an iterative process but initial conditions of the convergence assessment are not described. How many regions with their respective HCTs are required to begin with? How many ROIs are required to achieve convergence? What is the minimal and maximal number of ROIs to assure that convergence can be achieved?

We modified the following sentences in the **revised manuscript, in section 2.4 at page 8 and 10 to specifically address this important point:**

“To assess for statistical convergence of individual HCTs to a final IHCT, after processing 80 tiles (≈ 800 cells), Chi-squared distance was calculated by adding each new ROI tile with an empirically determined threshold to assess when the IHCT is converged. Once it's converged, the bone marrow NDC is completed and is represented by the generated IHCT. Otherwise, another ROI tile is extracted and analyzed interactively until convergence. Based on analysis of 500 individual patients' WSIs, we found that in most cases, IHCT convergence is reached after counting cells in 100-200 tiles for normal-diagnosed patients, 300-400 tiles in patients with a MDS-diagnosed patients and 400-500 tiles in patients with a AML-diagnosed patients (≈ 1000 -5000 cells) (Fig.5b).

The authors provided ranges of ROI and cell types for the IHCT convergence to be achieved, but it is unclear whether these numbers can be universally used (or observed) for aspirates from normal vs. abnormal (i.e containing abnormal cells) bone marrow. Why measuring convergence is even needed here?

There are no accepted or validated reference ranges for bone marrow cytology. Expert hematopathologists learn and accept cell count ranges based on years of training and experience, and this is considered the ground truth or reference standard. In terms of the number of cell categories that we used, this is well-established and based on international ICSH clinical practice guidelines: 1. LEE, S. -H., ERBER, W. N., PORWIT, A., TOMONAGA, M. & PETERSON, L. C. ICSH guidelines for the standardization of bone marrow specimens and reports. *Int J Lab Hematol* **30**, 349-364 (2008). The point of whether the convergence the IHCT happens in both normal and abnormal is an excellent one, and we thank the reviewer for bringing this to our attention. **We modified section 2.4 (as explained above) to show examples of IHCT convergence for both normal samples, and for samples with various abnormal hematological diagnoses (AML and MDS).** We use convergence here by chi-squared distance as a means to assess whether the cell counting algorithm may be stopped, as a stable, converged histogram would on average represent the distribution of cell types in that patient specimen.

BTW, why a simple histogram of cells from all regions across a WSI is not a good way of representing the cell frequencies, and why IHCT would be better than this histogram?

This is an excellent question, and perhaps we did not clarify this in the paper. An aspirate WSI contains only a relatively small number of regions suitable for cytology; these are well-spread, thin areas that are free of significant cellular overlap, overstaining and debris. These are the regions that are used by practicing hematopathologists throughout the world; we therefore designed our technology to reflect current clinical practice in hematopathology. The WSI regions that are not used for cytology (roughly 80% of the slide) are excluded because hematopathologists cannot accurately or reliably identify cells in these regions. Therefore, we could, and should not integrate suboptimal specimen regions in our technology, toward designing a real-world clinical diagnostic workflow support tool. **We added the following sentence as the last sentence of section 2.4 at page 10 of the revised manuscript:**

“Looking for only a small number of tiles in this approach is time and computationally efficient in comparison to analyzing all regions across a WSI, similar to real-world clinical practice.”

5) What is the value of MSE measured between NDCs by expert 1 and 2? Can the authors provide a quantitative evidence that the human operator-performed NDC is subject to increased intra-category variability?

We appreciate the concern to assess human operator variability in bone marrow counts. There are some studies assessing this in the past in an attempt to establish small reference ranges (1. Bain, B. J. The bone marrow aspirate of healthy subjects. *Brit J Haematol* **94**, 206–209 (1996). As our study was retrospective, many of the NDC were performed historically and therefore could not be feasibly reassessed within the scope of this study by multiple hematopathologists. Furthermore, our IHCT assessed more cytological object categories than the manual NDC. To avoid confusion, we have removed reference to historic NDC from the manuscript, and we now show data for hematopathologist concordance in evaluating the performance of the IHCT, which we feel is a more accurate metric of our model performance.. **Therefore, section 2.5 has been accordingly updated at page 12 of the revised manuscript.**

6) I doubt section 2.5 can be called clinical validation of model performance. What the authors did here is the identification of differences between human counts vs. IHCT yielded by AI by MSE. A clinical validation would include statistical tests showing that results of clinical decisions made based on IHCT are similar/concordant or same when compared to those based on manual NDC. At this point, it is also unclear how useful IHCT would be comparing to NDCs clinically.

Thank you for raising this important point; we agree that clinical validation may not be the most accurate way to describe this comparison study. We now rename this section and inherent experiments as **Concordance Between Hematopathologists and Model Performance**. The inference that an IHCT can be used clinically, or to influence clinical decisions may be inferred based on the concordance between hematopathologists IHCT in this section, however we do not draw this conclusion in the revised manuscript. For example, if a manual NDC identified 20% blasts, a clinical decision is made to treat an acute leukemia; therefore, if the IHCT shows the same or similar % of blasts; it would likewise influence the same clinical decision. We appreciate this comment however, and avoid making any specific conclusions regarding utilizing the IHCT for clinical-decision making in the manuscript.

7) The authors said that they used 100 samples (1 sample per patient) in their experiment. What kind of samples were used here (say how many from normal bone marrows vs., abnormal ones) ? What MSEs per group would be like when this potentially non-uniform sample of aspirates is broken down into clinically meaningful groups?

We thank the reviewer for this comment. As discussed in detail above, the categories analyzed in these historical human-performed NDC were less than those analyzed by our model, and these were historic counts that could not be feasibly re-assessed within the scope of our study. As mentioned above in (5), we now rename this section and present data evaluating concordance between hematopathologists and model performance, and between individual hematopathologists. We feel this approach more clearly assesses model performance. **The text and diagram has been updated at section 2.5 at page 12 accordingly.**

8) Another problem here is that the authors used manual ground truth to train and validate their AI. However, manual cell labeling is inaccurate and has high intra and inter-rater variability. A more appropriate would be to use immunohistochemistry to label cells for training, validation and testing purposes. An example of such approach is shown here: <https://www.ncbi.nlm.nih.gov/pmc/articles/PMC7177428/> Since immunohistochemistry (IHC) is likely to be more accurate for cell labeling, the authors should show classification results by their AI on IHC labeled cells or aspirates as the ground truth.

Thank you for the comment. With due respect, to clarify from the perspective and knowledge of the numerous hematopathologists and hematologists involved in this work, immunohistochemistry (IHC) or immunofluorescence using flow cytometry are not the reference standard for aspirate cell distributions in clinical medicine. The reference standard for bone marrow cell distributions is **expert-pathologist cell counts and manual annotations**. This is documented in the ICSH International Guidelines: 1.LEE, S. -H., ERBER, W. N., PORWIT, A., TOMONAGA, M. & PETERSON, L. C. ICSH guidelines for the standardization of bone marrow specimens and reports. *Int J Lab Hematol* **30**, 349 364 (2008); 2. .Torlakovic, E. E. *et al.* ICSH guidelines for the standardization of bone marrow immunohistochemistry. *Int J Lab Hematol* **37**, 431 449 (2015). Therefore, pathologist annotation for bone marrow cell counts is considered the ground truth; this is well-established in the field, and it is the benchmark for clinical diagnosis and patient care throughout the world. We have mentioned this several times, with appropriate references in the revised manuscript, **including section 4.2 on page 14.**

Bone marrow IHC is performed on solid tissue bone marrow specimens that complement cytology but do not provide a reference standard. IHC is an ancillary diagnostic test that is to assist pathologists in identifying the lineage or maturational stage of cells in bone marrow solid tissue specimens. There is no data to support its clinical use in correlating or supporting manual aspirate cell counts, which is a distinct and separate test. In regards to using flow cytometry (immunofluorescence) as in the reference cited by the reviewer, this an ancillary test to morphology that provides information about cell lineage and maturational stage to assist in pathology diagnosis. It is never used clinically to calculate cell distributions, and in fact, this would contravene best practice guidelines. Therefore, with respect, the reference cited by the reviewer does not accurately reflect the clinical use of bone marrow IHC. We appreciate this may be different from the research domain. We therefore feel that the cell annotation and assessment of our model by a team of 4 expert hematopathologists and hematologists with 5-35 years of pathology experience is more than

sufficient as a reference standard for our work. This again would agree with international best-practice clinical guidelines.

To further clarify this point, we added and highlighted the following sentence in **section 2.3 at page 7 of the revised manuscript**:

“During the first and second training iterations (before implementing active learning), new ROI tiles were fully annotated manually by an expert hematopathologist to train our YOLO model, 719 number of tiles representing 32 number of WSI were used for full cell annotation. From the third iteration onward, our active learning approach was employed to annotate 2766 new tiles representing an additional 74 WSIs, which were validated by our team of 4 expert hematopathologists and hematologists with 5-35 years of pathology experience (Table 5 and Supplementary Table S2).”

9) Active learning is not necessarily a new approach in digital pathology. It has been or recommended for use before. See these example papers:

https://link.springer.com/chapter/10.1007/978-3-030-23937-4_3

<https://onlinelibrary.wiley.com/doi/full/10.1002/path.5331>

<https://pubmed.ncbi.nlm.nih.gov/32758706/>

Inaccurate or unproven statements: (1) Abstract: “HCT has potential to revolutionize hematopathology diagnostic workflows, leading to more cost-effective, accurate diagnosis and opening the door to precision medicine”. (2) Pg.2: “YOLO has not been applied to medical domain problems such as pathology”. Here are examples where YOLO was used in digital pathology.

<https://pubmed.ncbi.nlm.nih.gov/31476576/>,

<https://www.osapublishing.org/osac/fulltext.cfm?uri=osac-4-2-323&id=446861> and there are more.

Thank you for bringing this to our attention, and we agree.

(1) We now revise the abstract with the following statement:

“HCT has potential to eventually support more cost-effective and efficient hematopathology diagnostic workflows, and support AI-enabled computational pathology.”

(2) Also, the introduction has been modified by adding the following sentence at **page 2 of the revised manuscript**:

“For example in [36], YOLO has been applied to assess the cell types in bone marrow smears. However, only 7 cell types have been considered in that study. Moreover, the tiles need to be selected manually by the user.”

Reviewer #3 (Remarks to the Author):

The authors create a tool for automated evaluation of bone marrow aspirate smears using deep learning methodologies. In their method, they identify appropriate regions of interest and use the YOLO model to classify cells in the smears. Their method was improved through active learning through iterative review of model output by hematopathologists. They ultimately validated their model through the review of 2 additional hematopathologists reviewing 100 patient samples.

Overall, this paper describes an exciting new tool for automated bone marrow aspirate differential that represents a great opportunity for AI to assist in and streamline diagnoses from bone marrow samples. The authors have dealt with one of the major challenges of bone marrow aspirate evaluation, the identification of appropriate regions of interest, and have demonstrated the benefits of active learning in improving performance of the model.

Critiques:

1) The authors do not address either the types of disease entities used in this study or how different morphologic abnormalities might affect its performance. The numbers of each disease reviewed are not described. For example, how many smears represented myelodysplastic syndrome (MDS) and did abnormal morphology seen in MDS affect precision? I would expect it to. Furthermore, how many pathologist/AI discrepancies resulted from low numbers of a cell type (as mentioned in the discussion). E.g., might some of the discrepancy reflect enhanced diagnostic ability through counting of so many cells (and detection of very rare blasts not seen by the human, for example)?

Thank you for the very helpful comment. The individual patient-level confusion matrices have been provided for **Normal, MDS and Leukemia cases in Figure S2 at page 23 of the revised manuscript**. Also, the number of cases analyzed in our dataset sample for each diagnosis can be found in **Table 5 at page 15 of the revised manuscript**. In addition we have incorporated new data from 27 additional patient WSIs in our evaluation there, which includes new 12 Normal, 12 MDS and 3 leukemia cases, where 10 tiles from each WSIs were sampled and assessed by a hematopathologist for the cell detection and classification model performance, now incorporated into **Table 5 and add the following sentence in section 2.2 page 5 of the revised manuscript**:

Model performance on the specific individual diagnostic categories of normal, MDS and leukemia can be found in Supplementary Fig.S2."

2) **How did the model handle other/unidentifiable cells (e.g. cells with very abnormal morphology).**

We thank our clinical colleague for the excellent and important question. As you well know, there are always some cells even the most experienced pathologists cannot identify with absolute confidence. We addressed specifically as follows:

1. During the annotation process, where cells that were not identified with absolute confidence by our hematopathologists or hemtologists were labelled as "other". This is similar to clinical practice, as you know. Therefore the model would be trained not to assign these to any 1 specific category. While there are clear weaknesses in such an approach, due to the requirements of YOLO model training, leaving cells without an annotation was not possible.
2. The model was able to predict cells as belonging to the category "other" if it did not assign a given confidence threshold to another cell. While there are caveats to this approach, in the context of an early prototype, we felt this approach both reflected clinical practice and allowed for model evaluation in handling complex or challenging cytology.

Thank you again for this excellent question. **We modified and highlighted the following sentence in section 2.2 at page 5:**

"Similar to clinical practice, objects that could not be classified with certainty by a hematopathologist were labeled as "other cells". Therefore the model would be trained not to assign these to any specific category.

While there are clear weaknesses in such an approach, due to the requirements of YOLO model training, leaving cells without an annotation was not possible.”

3) The authors raise the issue of computational time in the introduction. They should describe the run time and whether or not it is feasible to implement in clinical practice.

Thank you for the great comment. **The following sentence has been added to the manuscript in section 3 at page 14 of the revised manuscript:**

*“Regarding the execution time in model deployment and production phase, it took approximately 4 minutes to generate the appropriate tiles (includes reading a digital WSI from disk, creating the tiles with a size of 512*512 pixels, and applying the proposed ROI detection model). Consequently, the whole process to examine each tile takes about 30 milliseconds. For cell detection and classification, 50 milliseconds on average take for both detection and classification in each tile. As in most cases, the IHCT is converged in almost 400 to 500 tiles, the whole process of generating the IHCT took approximately 5 minutes.”*

4) There is a caveat in the discussion that is difficult to follow: “Finally, when compared to manual NDC in 100 patients, the model performed well...” This part seems to suggest the possibility of interobserver variability; though a high degree of interobserver variability would be unexpected in the identification of neutrophils in particular. It would be best for the authors to write these two sentences more clearly as they are hard to follow. But also, is it possible that because neutrophils are so frequent in aspirate smears, that it might affect the variance in counts, contributing to the observations described (or maybe not, just a thought).

Thank you again - to avoid confusion, the comparison between the manual NDC and the model has been removed from the revised manuscript, as it was retrospective, and only 10 cell types were considered at that study, but our proposed model detects and classifies 19 different cell types. We now rename this section, and present data evaluating concordance between individual hematopathologists and model performance, and between individual hematopathologists. We feel this approach more clearly assesses model performance. **The text and diagram has been updated at section 2.5 at page 12 accordingly:**

“For comprehensive clinical evaluation of our end-to-end model, performance was evaluated by two additional hematopathologists who were not involved in cell labeling. Both hematopathologists showed high concordance with model performance, with mAP > 90% overall cell and object types (Fig. 6 for details). Additionally, the Cohen’s Kappa has been calculated both between the hematopathologists and model, and within experts as well; 0.97 and 0.98 for expert 1 and expert 2, respectively and 0.99 between both experts.”

Reviewers' comments:

Reviewer #1 (Remarks to the Author):

The authors have followed reviewer's comment to improve their manuscript as the rebuttal letter. I agree in principle for all responses given by authors. However, I still hope that the author can further clarify on the two issues as following.

1. Original comments: The term "Histogram of Cell Types" is misleading due to a specific term of "Image histogram" in computer vision domain. In fact, it just a distribution of each cell type. Please revise the name of your paper.

Authors' response: Thank you for the comment. We feel that for the purposes of this work, the terms distribution and histogram are overlapping and analogous terms. As this paper is clinically-oriented, rather than oriented at the ML community, the term histogram conveys the purpose of the work clearly to the clinical/pathologist audience, i.e., we are automating a bone marrow manual nucleated differential count which is a distribution of cells in various categories as a histogram. Similarly, a manual nucleated differential count can be called a histogram in clinical medicine. We therefore feel that "Histogram of Cell Types" is the most accurate and meaningful way to convey our technology and results to our clinical colleagues. This should not be confusing to our readers, as the primary audience in this journal will be clinical pathologists versus deep learning / computer vision specialists. As well, it is generally understood that a "histogram" is a quantification of counting discretized measurements. We also added in Introduction: "A histogram is generally a representation of a distribution, a very old graphical technique to count discrete values [37]." My additional opinion: I am an expert in deep learning and statistics, and I still considered that term of this manuscript, "Histogram of Cell Types", might lead readers misunderstanding. I strongly recommend to replace this term to "Proportion of Cell Type". I suggested to seek an external statistical expert for additional reviews.

2. Original comments: There were 3 iterations for data collection in this study. Please provided the performance details of each iteration. For example, how many samples (especially in the sample of rare cell type) were collected from first, second, and third iterations for training and validation, and the performance should be provided. Importantly, is the validation set vary in each iteration?

Authors' response: Thank you very much for the comment. The details of each iteration have been provided inside the manuscript in Tables 3 page 4. Regarding the second question, yes, in each iteration a new Training, and Test-Validation sets were created and the model trained and evaluated on them. Regarding "samples of rare cell types"; we randomly selected patient WSI samples from our starting dataset. We assume by rare cell types, you mean cells such as blasts, megakaryocytes or histiocytes. As can be seen in Table 3, these were all sampled proportionally across iterations.

My additional opinion: Because the accuracy in different validation set is not comparable, I recommend to create an equal validation set for all iterations, which may be helpful for understanding the improving by iterations. I think the experiment I requested should at least be placed in the supplementary file.

Reviewer #2 (Remarks to the Author):

Thank you for addressing my comments. a couple of minor suggestions.

- 1) Please make sure the mAP abbreviation is resolved in the abstract or in the text where it first appears.
- 2) Describe which of the query strategies of active learning (AL) scheme was used in your work. If all the wrongly classified cells were re-labeled during a single AL iteration, please say so in the manuscript.

Reviewer #3 (Remarks to the Author):

The authors create a tool for automated evaluation of bone marrow aspirate smears using deep learning methodologies. In their method, they identify appropriate regions of interest and use the YOLO model to classify cells in the smears. Their method was improved through active learning through iterative review of model output by hematopathologists. Overall, this paper describes an exciting new tool for automated bone marrow aspirate differential that represents a great opportunity for AI to assist in and streamline diagnoses from bone marrow samples. The authors have dealt with one of the major challenges of bone marrow aspirate evaluation, the identification of appropriate regions of interest, and have demonstrated the benefits of active learning in improving performance of the model. The authors highlight the several ways that this technology can transform hematopathology practice and potentially improve diagnostic abilities (e.g. in the detection of MRD). This method represents a substantial advancement relative to prior methods that required a human operator to identify regions of interest and also only identified a smaller number of cell types. In the revised manuscript, the authors have addressed most of my major concerns from the initial review. However, the additional data provided raises new questions. I have a few remaining and additional critiques:

- 1) My biggest concern: How should the pathologist deal with the fact that 24% of blasts are misclassified in the MDS cases? Some discussion of how to resolve this major problem with the model, beyond explaining that human pathologists also have trouble with this, would be helpful. The benefit of the human is that they can recognize the ambiguity and decide on the final classification of each cell using additional information (flow cytometry, IHC, clinical impression) whereas it seems that the model is simply classifying 14% of blasts as lymphocytes and another 4% as erythroblasts. Quantification of blasts in MDS is one of the most important functions of the bone marrow aspirate smear and this issue deserves more attention. I find the confusion matrix provided in the main paper to be misleading as it seems to represent mostly normal cases. I think the fact that accuracy, particularly with regard to blast count, is lost in abnormal cases should be better acknowledged.
- 2) Can confusion matrices for the other disease entities evaluated be provided?
- 3) I wish that the authors had also discussed the contribution of dysplastic morphology (e.g. in MDS) to the accuracy of the model. It seems, from Fig S2, that there was some increase in misclassification (aside from the blast issue described above) in MDS compared to other smear types. Whether or not these misclassifications represent a major problem should have been discussed (I think that most are in cell classes that are not of major clinical significance; eg megakaryocyte nuclei, platelet clumps).
- 4) Minor issue: The authors frequently refer to "leukemia" cases without more detail about the type. I assume they mean acute leukemia? If it was chronic lymphocytic leukemia or chronic myeloid leukemia, this should be clarified as the cell types would differ. It's especially confusing in the tables/figures.

Reviewers' comments:

Reviewer #1 (Remarks to the Author):

The authors have followed reviewer's comment to improve their manuscript as the rebuttal letter. I agree in principle for all responses given by authors. However, I still hope that the author can further clarify on the two issues as following.

1. Original comments: The term "Histogram of Cell Types" is misleading due to a specific term of "Image histogram" in computer vision domain. In fact, it just a distribution of each cell type. Please revise the name of your paper.

Authors' response: Thank you for the comment. We feel that for the purposes of this work, the terms distribution and histogram are overlapping and analogous terms. As this paper is clinically-oriented, rather than oriented at the ML community, the term histogram conveys the purpose of the work clearly to the clinical/pathologist audience, i.e., we are automating a bone marrow manual nucleated differential count which is a distribution of cells in various categories as a histogram. Similarly, a manual nucleated differential count can be called a histogram in clinical medicine. We therefore feel that "Histogram of Cell Types" is the most accurate and meaningful way to convey our technology and results to our clinical colleagues. This should not be confusing to our readers, as the primary audience in this journal will be clinical pathologists versus deep learning / computer vision specialists. As well, it is generally understood that a "histogram" is a quantification of counting discretized measurements. We also added in Introduction: "A histogram is generally a representation of a distribution, a very old graphical technique to count discrete values [37]." **My additional opinion: I am an expert in deep learning and statistics, and I still considered that term of this manuscript, "Histogram of Cell Types", might lead readers misunderstanding. I strongly recommend to replace this term to "Proportion of Cell Type". I suggested to seek an external statistical expert for additional reviews.**

Response to 1: We thank the reviewer again for the excellent and thoughtful comments, which have supported a considerably improved manuscript throughout the review process. With regard to the title of the paper, we appreciate your continued concern for reader misunderstanding. We agree, if this work was submitted to target a computer vision audience, this could be a concern. However, we are primarily targeting a clinical pathologist audience, where the term "*Histogram of Cell Types*" conveys the clinical utility of the information representation presented here to this demographic. Specifically, the use of histograms for cell counts is a standard and established part of hematopathology diagnostic workflows. Therefore, we feel it is important to analogously term our bone marrow cell count representation as a *histogram* specifically for clarity to this pathologist audience. We hope the reviewer will consider this perspective, and again, we sincerely thank the reviewer for the very excellent and thoughtful comments regarding this work.

2. Original comments: There were 3 iterations for data collection in this study. Please provided the performance details of each iteration. For example, how many samples (especially in the sample of rare cell type) were collected from first, second, and third iterations for training and validation, and the performance should be provided. Importantly, is the validation set vary in each iteration?

Authors' response: Thank you very much for the comment. The details of each iteration have been provided inside the manuscript in Tables 3 page 4. Regarding the second question, yes, in each iteration a new Training, and Test-Validation sets were created and the model trained and evaluated on them. Regarding "samples of rare cell types"; we randomly selected patient WSI samples from our starting dataset. We assume by rare cell types, you mean cells such as blasts, megakaryocytes or histiocytes. As can be seen in Table 3, these were all sampled proportionally across iterations.

My additional opinion: Because the accuracy in different validation set is not comparable, I recommend to create an equal validation set for all iterations, which may be helpful for understanding the improving by iterations. I think the experiment I requested should at least be placed in the supplementary file.

Response to 2: Thank you for bringing this to our attention. Specifically, we now include new data demonstrating model performance over all iterations with a single validation set, as described in **supplementary Table S3, page 2 of supplementary data, and referenced on page 7 of the revised manuscript**. As can be seen, there was also improvement in model performance with a single validation dataset.

Reviewer #2 (Remarks to the Author):

Thank you for addressing my comments. a couple of minor suggestions.

1) Please make sure the mAP abbreviation is resolved in the abstract or in the text where it first appears.

Response to 1: Thank you for the comment. The following sentence has been updated in the introduction of the **revised manuscript on page 3 and also in the abstract**:

"Our approach shows cross-validation accuracy of 0.97 and precision of 0.90 in ROI detection (selecting appropriate tiles), and mAP (mean Average Precision) of 0.75 and average F1-score of 0.78 for detecting and classifying 16 key cellular and non-cellular objects in aspirate WSIs."

2) Describe which of the query strategies of active learning (AL) scheme was used in your work. If all the wrongly classified cells were re-labeled during a single AL iteration, please say so in the manuscript.

Response to 2: Thank you for the important comment. The following sentence has been added to **section 2.3 at page 7 of the revised manuscript**:

“In this way, the expected error reduction (EER) approach of active learning was used; at each iteration, wrongly classified cells were re-labeled and added to the current training dataset to train the model in the next iteration.

Reviewer #3 (Remarks to the Author):

The authors create a tool for automated evaluation of bone marrow aspirate smears using deep learning methodologies. In their method, they identify appropriate regions of interest and use the YOLO model to classify cells in the smears. Their method was improved through active learning through iterative review of model output by hematopathologists. Overall, this paper describes an exciting new tool for automated bone marrow aspirate differential that represents a great opportunity for AI to assist in and streamline diagnoses from bone marrow samples. The authors have dealt with one of the major challenges of bone marrow aspirate evaluation, the identification of appropriate regions of interest, and have demonstrated the benefits of active learning in improving performance of the model. The authors highlight the several ways that this technology can transform hematopathology practice and potentially improve diagnostic abilities (e.g. in the detection of MRD). This method

represents a substantial advancement relative to prior methods that required a human operator to identify regions of interest and also only identified a smaller number of cell types. In the revised manuscript, the authors have addressed most of my major concerns from the initial review. However, the additional data provided raises new questions. I have a few remaining and additional critiques:

1) My biggest concern: How should the pathologist deal with the fact that 24% of blasts are misclassified in the MDS cases? Some discussion of how to resolve this major problem with the model, beyond explaining that human pathologists also have trouble with this, would be helpful. The benefit of the human is that they can recognize the ambiguity and decide on the final classification of each cell using additional information (flow cytometry, IHC, clinical impression) whereas it seems that the model is simply classifying 14% of blasts as lymphocytes and another 4% as erythroblasts. Quantification of blasts in MDS is one of the most important functions of the bone marrow aspirate smear and this issue deserves more attention. I find the confusion matrix provided in the main paper to be misleading as it seems to represent mostly normal cases. I think the fact that accuracy, particularly with regard to blast count, is lost in abnormal cases should be better acknowledged.

Response to points 1 and 3 from the reviewer: These are excellent and clinically relevant points. We now add significant new text to address and clearly acknowledge these issues as recommended:

This is highlighted in the results section of the revised manuscript, page 5, as follows:

“Cell types such as “blasts” and “lymphocytes”, which may show overlapping morphological features, also showed lower model performance in accuracy, similar to expert human hematopathologists. Model performance in the specific individual diagnostic categories of normal, MDS, acute leukemia, plasma cell neoplasm and lymphoproliferative disorder can be found in Supplementary Fig. S2.”

and these important points are addressed in detail in discussion section of the revised manuscript page 13, as follows:

Specifically, “blasts” and “lymphocytes” showed overlap in classification and lower accuracy by our model (Fig S2), which is a similar problem to human hematopathologists. A proportion of “Plasma cells” were also misclassified as “erythroblasts” in cases diagnosed as “plasma cell neoplasm” (Fig S2). This may reflect biases in model performance, or alternatively, may be a function of the overlapping cytological features in these cell types which are often confused in clinical practice, specifically in MDS where dysplasia renders morphology challenging. We acknowledge this as a weakness in our model, one that is somewhat mitigated in real-world clinical practice by expert human hematopathologists using integrated, semantic interpretation of multiple ancillary data modalities, such as flow cytometry, molecular studies and clinical findings. As an early prototype, this problem may be potentially addressed in future work by incorporating additional training data, as well as multi-model ML-analyzed datasets and active learning approaches. It is critical to emphasize our ML technology would be intended to support pathologists and expedite workflows, requiring substantive human oversight and rigorous clinical validation, especially where blast counts represent critical diagnostic cutoffs in diagnosing MDS and acute leukemias.

Morphology in MDS poses a challenge for even the most experienced pathologists, with high inter-observer variability, as reflected in our model performance of MDS cases (Figure S2b). This is a potential problem in a model that uses discrete class probability assignments for individual cells, where there still may be significant intra-class heterogeneity. Future iterations of our model may use approaches such as deep feature extraction from YOLO and dimensionality reduction to explore unsupervised relationships between cells in each class, ex, dysplasia within neutrophils, which may assist pathologists in interpreting cell subsets in cases with morphological dysplasia. This type of approach would yield additional information that goes beyond simple class probability assignment, allowing pathologists to understand and visualize cytological relationships learned by a model.

With regard to the concern about the confusion matrix in **Figure 2b of the main paper** being possibly misleading as to the normal/abnormal case balance, we remind the reviewer that the cases analyzed and presented here represent 10 diagnostic classes

and 106 patient WSI, as shown in **Table 5 page 15 and previously stated on page 5 of the revised manuscript**. While the “normal cases” have the largest number in a single diagnostic category, in fact, they represent a minority of diagnostic tags, as the other 74 cases (majority) come from “abnormal” diagnosed cases in 9 additional diagnostic categories. We hope the reviewer will agree this is not misleading, and for further clarity, we have now highlighted this explicitly in **figure 2 and on page 5 of the revised manuscript**.

2) Can confusion matrices for the other disease entities evaluated be provided?

Thank you for the comment. The individual patient-level confusion matrices have been updated by now adding the additional diagnostic categories of plasma cell neoplasm and lymphoproliferative disorder in **Figure S2 page 3 of supplementary data, and referenced on page 5 of the revised manuscript**. As the reviewer is a clinical hematopathology colleague, we hope they will agree that our confusion matrix data now represents a broad spectrum of common normal and abnormal diagnostic categories that reflect daily hematopathology practice. We genuinely thank our expert clinical colleague for the excellent and thoughtful comments which we feel have resulted in a substantially improved manuscript. As mentioned, we also add detailed comment on the limitations of our model, and possible future strategies to mitigate this in the **discussion section on page 13 of the revised manuscript**.

3) I wish that the authors had also discussed the contribution of dysplastic morphology (e.g. in MDS) to the accuracy of the model. It seems, from Fig S2, that there was some increase in misclassification (aside from the blast issue described above) in MDS compared to other smear types. Whether or not these misclassifications represent a major problem should have been discussed (I think that most are in cell classes that are not of major clinical significance; eg megakaryocyte nuclei, platelet clumps).

Thank you and agreed; integrated in discussion with comments on cell misclassification in **response to comment 1 above, added to the discussion section on page 13 of the revised manuscript**.

4) Minor issue: The authors frequently refer to “leukemia” cases without more detail about the type. I assume they mean acute leukemia? If it was chronic lymphocytic leukemia or chronic myeloid leukemia, this should be clarified as the cell types would differ. It’s especially confusing in the tables/figures.

Yes, we indeed are referring to acute leukemia. We have now resolved this in the text and tables / legends in the **revised manuscript**.

REVIEWERS' COMMENTS:

Reviewer #1 (Remarks to the Author):

The authors have made proper language improvements to the manuscript. Thank you, I have nothing further to add.

Reviewer #3 (Remarks to the Author):

Thank you for responding to my critiques.

While the concerns about the model's accuracy in in the context of blast and plasma cell counts remain, I feel that the authors have sufficiently highlighted this issue in the discussion of the revised manuscript and have offered both plans for future improvements and ideas for how the tool, despite its limitations, can be used in conjunction with other tools, human oversight, etc.

Figure S2 still refers to "leukemia" in both the legend and the label. I recommend changing this to "acute leukemia" or "acute myeloid leukemia," whichever is more appropriate.

REVIEWERS' COMMENTS:

Reviewer #1 (Remarks to the Author):

The authors have made proper language improvements to the manuscript. Thank you, I have nothing further to add.

Response: We thank the reviewer for their valuable expert-content input throughout the review process.

Reviewer #3 (Remarks to the Author):

Thank you for responding to my critiques.

While the concerns about the model's accuracy in in the context of blast and plasma cell counts remain, I feel that the authors have sufficiently highlighted this issue in the discussion of the revised manuscript and have offered both plans for future improvements and ideas for how the tool, despite its limitations, can be used in conjunction with other tools, human oversight, etc.

Figure S2 still refers to "leukemia" in both the legend and the label. I recommend changing this to "acute leukemia" or "acute myeloid leukemia," whichever is more appropriate.

Response: We thank the reviewer for their valuable expert-content input throughout the review process. We now correct the legend to acute leukemia as indicated.